# Acidification, deoxygenation, nutrient and biomasses decline in a warming Mediterranean Sea

Marco Reale[1], Gianpiero Cossarini[1], Paolo Lazzari[1], Tomas Lovato[2], Giorgio Bolzon[1], Simona Masina[2], Cosimo Solidoro[1], Stefano Salon[1]

1. National Institute of Oceanography and Applied Geophysics - OGS, Trieste, Italy
2. Fondazione Centro Euro-Mediterraneo sui Cambiamenti Climatici, CMCC, Ocean Modeling and Data Assimilation Division, Bologna, Italy

*Correspondence to:* Marco Reale (mreale@inogs.it) and Stefano Salon (ssalon@inogs.it)

**Abstract.** The projected warming, nutrient decline, changes in net primary production, deoxygenation and acidification of the global ocean will affect marine ecosystems during the 21st century. Here the climate change-related impacts in the marine ecosystems of the Mediterranean Sea in the middle and at the end of the 21st century are assessed using high-resolution projections of the physical and biogeochemical state of the basin under the Representative Concentration Pathways (RCPs) 4.5 and 8.5. The analysis shows in both scenarios changes in the dissolved nutrient content of the euphotic and intermediate layers of the basin, net primary production, phytoplankton respiration and carbon stock (including phytoplankton, zooplankton, bacterial biomass and particulate organic matter). The projections also show a uniform surface and subsurface reduction in the oxygen concentration driven by the warming of the water column and by the increase in ecosystem respiration, and an acidification signal in the upper water column, linked to the increase in the dissolved inorganic carbon content of the water column due to $CO_2$ absorption from the atmosphere and the increase in respiration. The projected changes are stronger in the RCP8.5 (worst-case) scenario and, in particular, in the Eastern Mediterranean due to the limited influence of the exchanges in the Strait of Gibraltar in that part of the basin. On the other hand, the analysis of the projections under RCP4.5 emission scenario shows a tendency to recover the values observed at the beginning of the 21st century for several biogeochemical variables in the second half of the period. This result supports the idea - possibly based on the existence, in a system like the Mediterranean Sea, of a certain buffer capacity and renewal rate - that the implementation of policies of reducing $CO_2$ emission could be, indeed, effective and could contribute to the foundation of ocean sustainability science and policies.

## 1. Introduction

The Mediterranean Sea (Fig. 1) is a mid-latitude semi-enclosed basin surrounded by the continental areas of Southern Europe, Northern Africa and the Middle East. The basin is characterized by a thermohaline circulation composed of three distinctive cells. The first is an open cell associated with the inflow of the Atlantic Water (AW) at the Strait of Gibraltar (which undergoes a progressive increase in salinity due to evaporation becoming Modified Atlantic Water, or MAW) and the formation of Levantine Intermediate Water (LIW) in the Eastern basin (Lascaratos, 1993; Nittis and Lascaratos, 1998; Velaoras et al., 2019; Fach et al., 2021; Fedele et al., 2021). The other two are closed cells associated with deep water

formation processes occurring in the Gulf of Lion (located in the North Western Mediterranean, Fig.1; Somot et al., 2018 and reference therein) and in the Adriatic Sea (Fig. 1; Mantziafou and Lascaratos 2004, 2008; Schroeder et al., 2012 and references therein).

Future climate projections for the Mediterranean region based on different emission scenarios show, at the end of the 21st century, (i) a reduction in precipitation and a general warming of the area (e.g., Giorgi, 2006; Diffenbaugh et al., 2007; Giorgi and Lionello, 2008; Dubois et al., 2012; Lionello et al., 2012; Planton et al., 2012; Gualdi et al., 2013; MedEEC, 2020), (ii) a warming of seawater (Somot et al., 2006; Adloff et al., 2015; Soto-Navarro et al., 2020; MedECC, 2020), and (iii) a consistent weakening of the thermohaline circulation and an increase in the stratification index throughout the basin (Somot et al., 2006; Adloff et al., 2015; Soto-Navarro et al., 2020) and a further increase in frequency and severity of atmospheric and marine heat waves and drought (Galli et al., 2017; Darmaraki et al., 2019; Ibrahim et al., 2021; Mathbout et al., 2021). Conversely, the future evolution of sea surface salinity in the Mediterranean Sea and the sign of its change are still uncertain due to the role played by rivers and Strait of Gibraltar exchanges (Adloff et al., 2015; Soto-Navarro et al., 2020; MedECC, 2020). In general, the magnitude of the projected changes has been shown to be dependent on the adopted emission scenario (MedECC, 2020).

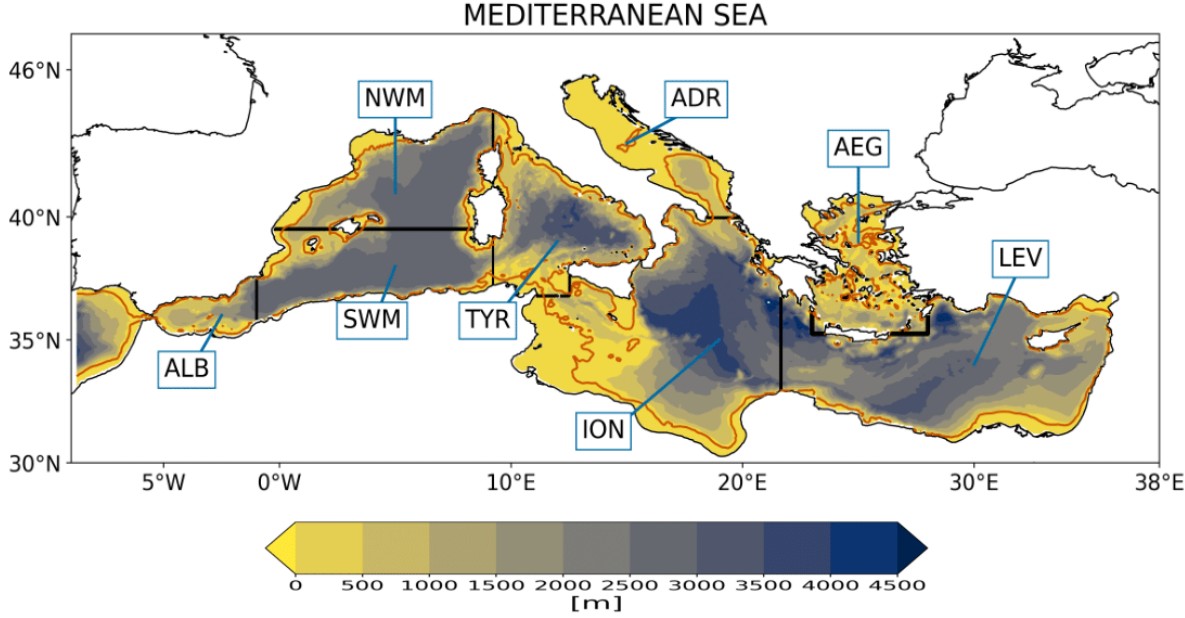

**Fig.1 Mediterranean Sea bathymetry (in m) and relative sub-basins considered in the analysis: Alboran Sea (ALB), North Western Mediterranean (NWM), South Western Mediterranean (SWM), Tyrrhenian (TYR), Adriatic Sea (ADR), Ionian Sea (ION), Aegean Sea (AEG), Levantine basin (LEV). The dark orange line marks the 200m isobath in the model domain. The domain boundary is set at longitude 8.8°W, westward of the Strait of Gibraltar.**

From a biogeochemical point of view, the Mediterranean Sea is considered as an oligotrophic (ultraoligotrophic in its Eastern part) basin (Bethoux et al., 1998; Moutin and Raimbault, 2002; Siokou-Frangou et al., 2010; Lazzari et al., 2012). It is characterized by low productivity levels and an east-west trophic gradient (Crise et al.,1999; D'Ortenzio and Ribera d'Alcala 2009; Lazzari et al., 2012) which results from the superposition of different mechanisms such as the biological pump, the estuarine inverse circulation, and the position of nitrate ($NO_3$) and phosphate ($PO_4$) sources (Crise et al., 1999; Crispi et al., 2001). The only exceptions to the oligotrophy of the basin are some areas (Gulf of Lion, Strait of Sicily,

Algerian coastlines, Southern Adriatic Sea, Ionian Sea, Aegean Sea and Rhodes Gyre) where strong vertical mixing and upwelling phenomena associated with air-sea interactions and wind stress forcing enrich the surface in nutrients, so favouring phytoplankton rapid growth (or bloom) mostly in the late winter-early spring period (D'Ortenzio and Ribera d'Alcala, 2009; Reale et al., 2020b). A proxy widely adopted to detect phytoplankton blooms is the surface concentration of chlorophyll-*a* (chl-a) that is characterized by relative high values in specific open sea/coastal areas, where it is linked to the physical forcing and river inflow (D'Ortenzio and Ribera d'Alcala, 2009; Lazzari et al., 2012; Herrmann et al., 2013; Auger et al., 2014; Richon et al., 2018; Di Biagio et al., 2019; Reale et al., 2020a). The open sea chl-*a* vertical dynamics follows a seasonal cycle with winter-early spring surface blooms, and summer onset of a deep chl-*a* maximum (DCM) which deepens from approximately 50 m in the Western areas to 100 m in the Eastern areas (e.g. Lazzari et al., 2012; Macias et al., 2014; Lavigne et al., 2015; Cossarini et al., 2021).

Due to the strong links between ocean/atmosphere dynamics and biogeochemical patterns, it has to be expected that future climate change will have relevant impacts on the biogeochemistry and, in turn, on the marine ecosystem dynamics of the Mediterranean Sea. In fact, all the projected changes for the region will likely affect the vertical mixing and reduce the nutrient supply into the euphotic layer of the Mediterranean Sea (e.g. Richon et al., 2019), which is essential for phytoplankton dynamics and productivity, with possible impacts on the biogeochemical carbon cycle and carbon dioxide ($CO_2$) exchange with the atmosphere (e.g., Lazzari et al., 2012; Cossarini et al., 2015; Canu et al., 2015; Solidoro et al., 2022).

An assessment of the effects of climate change on the biogeochemistry and marine ecosystem dynamics of the Mediterranean Sea has been considered in a number of previous studies based on different emission scenarios. Hermann et al. (2014) assessed the response of the pelagic planktonic ecosystem of the North Western Mediterranean to different emission scenarios and showed that, at end of the 21st century, the biogeochemical processes and marine ecosystem components should be very similar to those observed at the end of the 20th century, although quantitative differences might be observed, such as an increase in the bacteria growth, gross primary production and biomass of small-size phytoplankton group. Lazzari et al. (2014) found a negative change in the plankton biomass in response to the A1B emission scenario, resulting from an increase of productivity and community respiration. Benedetti et al. (2018), using environmental niche models and considering six physical simulations based on different emission scenarios (A2, A2-F, A2-RF, A2-ARF, A1B-ARF, B1-ARF; Adloff et al., 2015), projected, in response to climate change, a loss of copepods diversity throughout most of the surface layer of the Mediterranean Sea. On the other hand, Moullec et al. (2019) under RCP8.5 emission scenario found an increase/decrease in both phytoplankton biomass and net primary production by the end of the 21st century in the Eastern/Western Mediterranean Sea. Macias et al. (2015) showed that, under emission scenarios RCP4.5 and RCP8.5 and despite a significant observed warming trend, the mean integrated primary production rate in the entire basin will remain almost unchanged in the 21st century. However, they pointed out some peculiar spatial differences in the basin such as an increase in the oligotrophy of the Western basin due to a surface density decrease and an increase in net primary production in the Eastern basin due to the increased density. Richon et al. (2019) observed, under the A2 emission scenario (which is similar to the RCP8.5 emission scenario in terms of magnitude of the projected changes in the global mean temperature), an accumulation of nitrate in the basin and a reduction of 10% in net primary productivity by 2090, with a peak of 50% in specific areas (including the Aegean Sea). On the other hand, no tendencies in the phosphorus were observed. Pagès et al. (2020) showed, under emission scenario RCP8.5, a decline in the nutrient

concentration (stronger in $NO_3$ than $PO_4$) at the surface of the basin due to the increase in the vertical stratification and
pointed out that the Mediterranean Sea will become less productive (14% decrease in integrated primary production in
both Western and Eastern basins) and will be characterized by a reduction (22% in the Western basin and 38% in the
Eastern basin) in large phytoplankton species abundance in favor of small organisms. All these changes will mainly affect
the Western basin, while the Eastern basin will be less impacted (Pagès et al., 2020). Solidoro et al. (2022) discussed the
evolution of the carbon cycling, budgets and fluxes of the basin under the A2 scenario, highlighting an increase in the
trophodynamic carbon fluxes and showing, at the same time, that the increment in the plankton primary production will
be more than compensated by the increase in the ecosystem total respiration, which corresponds to a decrease of the total
living carbon and oxygen in the epipelagic layer. Moreover, Solidoro et al., (2022) also projected an increase of dissolved
inorganic carbon (DIC) pool and quantified for the first time the related acidification of the basin, a process that might
significantly alter the Mediterranean ecosystems (Zunino et al., 2017; 2019) and their capability to sustain ecosystem
services (Zunino et al., 2021).
All the above-mentioned works demonstrate that the dynamics of the marine ecosystem may be affected directly and
indirectly by climate change and the magnitude of their response is dependent on the emission scenario adopted. Different
levels of warming, acidification and changes in the vertical distribution of the oxygen, nutrient concentration and net
primary production related to water column stratification are all potential marine stressors affecting marine organisms
and ecosystem dynamics (see Kwiatkowski et al., 2020 for a review about the synergistic effects among potential marine
stressors).
A proper simulation of these marine stressors and related impacts require the adoption of suitable horizontal and vertical
resolutions. In fact, it has been shown that meso and submesoscale structures of the Mediterranean circulation influence
indeed the biogeochemical dynamics of many areas of the basin (Moutin and Prieur, 2012; Richon et al., 2019), while the
vertical resolution affects the features of the simulated stratification and subsurface ventilation pathways (see
Kwiatkowski et al., 2020 and reference therein for a review).
These considerations emphasize the importance of providing eddy-resolving future projections of the Mediterranean Sea
biogeochemistry under different emission scenarios. In fact, although observational and modeling studies have been
already carried out in the recent period to assess the importance of the mesoscale dynamics on the physical and
biogeochemical state of limited areas of the Mediterranean Sea (e.g. Hermann et al., 2008; Moutin and Prieur, 2012;
Guyennon et al., 2015; Ramirez-Romero et al., 2020), long-term eddy-resolving biogeochemical projections under
different emission scenarios, to the best of the authors' knowledge, have not been analyzed so far in the region. Such
projections might be used in future studies specifically focused on the analysis of climate change impact on specific
organisms, habitats and/or local areas.
Therefore, here climate change-related impacts in the marine ecosystems of the Mediterranean Sea in the middle and at
the end of the 21st century are assessed using eddy-resolving projections of the physical and biogeochemical state of the
basin under emission scenarios RCP4.5 and RCP8.5. These projections are derived from the offline coupling between the
physical model MFS16 (Mediterranean Forecasting System at 1/16°; Oddo et al., 2009) and the transport-reaction model
OGSTM-BFM (OGS Transport Model-Biogeochemical Flux Model; Lazzari et al., 2012). The analysis focuses on 21st
century projected changes of dissolved nutrients and oxygen, net primary production, respiration, living/non-living
organic matter, plankton and bacterial biomass, and particulate organic matter (POC). Moreover, the response of the basin
to the increasing atmospheric $CO_2$ concentrations is thoroughly investigated. The projected changes are also correlated
with changes in the physical forcing in the region.
The article is organized as follows: the MFS16-OGSTM-BFM system along with the physical forcing used to drive the
biogeochemical scenarios, initial and boundary conditions and numerical experiments are described in Section 2. Section
3 discusses the projected changes in climate change-related impacts in the marine ecosystems of the Mediterranean basin.
Finally, Section 4 summarizes and discusses the results of this work, together with their uncertainties, paving the way for
possible future research avenues.

**1.  Data and Methods**

The biogeochemical projections of the Mediterranean Sea state during the 21st century have been produced by driving
the transport-reaction model OGSTM-BFM (Lazzari et al., 2012) with the 3D outputs of the physical model MFS16
(Oddo et al., 2009) through an off-line coupling. In fact, the physical model MFS16 supplies to the OGSTM-BFM the
temporal evolution of daily horizontal and vertical current velocities, vertical eddy diffusivity, potential temperature,
salinity, and surface data for solar shortwave irradiance and wind stress. The resulting transport processes affecting the
concentration of biogeochemical tracers (advection, vertical diffusion and sinking) are computed by OGSTM, which is a
modified version of the OPA tracer model (Océan PArallélisé, Foujols et al., 2000). The temporal evolution of
biogeochemical processes is computed by the Biogeochemical Flux Model (BFM; Vichi et al., 2015).

**2.1. The MFS16 physical model**

MFS16 is the Mediterranean configuration of the NEMO modelling system (Nucleus for European Modelling of the
Ocean; Madec, 2008; see also http://www.nemo-ocean.eu, version 3.4) and constitutes the climate implementation of the
Mediterranean Ocean Forecasting System (Oddo et al., 2009; Lovato et al., 2013).

The original MFS16 domain covers the whole Mediterranean Sea and part of the neighboring Atlantic Ocean region with
a horizontal grid resolution of 1/16º (~6.5 km) and 72 unevenly spaced vertical levels (ranging from 3 m at the surface
down to 600 m in the deeper layers, see Lovato et al., 2013). The model computes the air-sea fluxes of water, momentum
and heat using specific bulk formulae tuned for the Mediterranean Sea (Oddo et al., 2009) applied to the atmospheric
fields obtained from the atmosphere-ocean general circulation model CMCC-CM (CMCC-Coupled model; Scoccimarro
et al., 2011).

The open boundary conditions in the Atlantic region for the physical variables (zonal/meridional component of current
velocity, sea surface height, temperature and salinity) were derived from the ocean component of the CMCC-CM coupled
model, while the riverine freshwater discharges and fluxes in the Dardanelles Strait were provided by the hydrological
component of the same coupled model (Gualdi et al., 2013). The initial conditions of the Mediterranean Sea were obtained
from the gridded temperature and salinity data produced by the SeaDataNet infrastructure (http://www.seadatanet.org/).
The model was initially spun-up for 25 years under present climate conditions and then scenario simulations were
performed over the 2005-2100 period.
**2.2. The OGSTM-BFM transport-reaction model**
The OGSTM-BFM transport-reaction model is based on the coupling of a transport model (OGSTM) based on the OPA
system (Foujols et al., 2000) and the BFM biogeochemical reactor. OGSTM-BFM is fully described in Lazzari et al.
(2012, 2016), where it was used to simulate chl-*a*, primary production and nutrient dynamics of the Mediterranean Sea
for the 1998-2004 period.
The OGSTM transport model resolves the advection, vertical diffusion and the sinking terms of the biogeochemical
tracers. The temporal scheme of OGSTM is an explicit forward temporal scheme for the advection and horizontal
diffusion terms, whereas an implicit time scheme is adopted for the vertical diffusion. The BFM biogeochemical reactor
considers co-occurring effects of multi-nutrient interactions and energy/material fluxes through the classical food chain
and the microbial food web which are both very important in the Mediterranean Sea (Bethoux et al., 1998). BFM has
been extensively applied to the studies of the dynamics of dissolved nutrients, chl-*a* and net primary production in the
Mediterranean Sea (Lazzari et al., 2012; 2016; Di Biagio et al., 2019; Reale et al., 2020a), marine carbon sequestration
and alkalinity (Canu et al., 2015; Cossarini et al., 2015; Butenschön et al., 2021), impacts of climate change on the
biogeochemical dynamics of marine ecosystems (Lazzari et al., 2014; Lamon et al., 2014; Solidoro et al., 2022), influence
of large-scale atmospheric circulation patterns on nutrient dynamics (Reale et al., 2020b) and operational short-term
forecasts for the Mediterranean Sea biogeochemistry (Teruzzi et al. 2018; 2019; Salon et al., 2019). The version adopted
here is the v5.
The model simulates the biogeochemical cycles of carbon, nitrogen, phosphorus and silicon through dissolved forms and
living organic and non-living organic compartments (labile, semi-labile and semi-refractory organic matter). Moreover,
it presently includes nine plankton functional types (PFTs), meant to be representative of diatoms, flagellates,
picophytoplankton, dinoflagellates, carnivorous and omnivorous mesozooplankton, bacteria, heterotrophic
nanoflagellates and microzooplankton. It also simulates the carbonate system dynamics, by solving the set of physico-
chemical equilibria related to total alkalinity (ALK) and dissolved inorganic carbon (DIC) chemical reactions (Cossarini
et al., 2015). ALK variability is driven by processes that alter the ion concentration in seawater (nitrification,
denitrification, uptake and release of nitrate, ammonia and phosphate by plankton cells, and precipitation and dissolution
of carbonate calcium-$CaCO_3$, see Wolf-Gladrow et al., 2007). DIC dynamics are driven by biological processes
(photosynthesis and respiration, precipitation and dissolution of $CaCO_3$) and physical processes ($CO_2$ exchanges at the
air-sea interface and, as for all the other biogeochemical tracers, dilution-concentration due to evaporation minus
precipitation processes).
**2.3. Initial and boundary conditions for the biogeochemistry**
Boundary conditions are adopted to represent the external supply of biogeochemical tracers and properties from the Strait
of Gibraltar and the rivers into the Mediterranean basin. The exchanges of nutrients and other biogeochemical tracers in

the Strait of Gibraltar are achieved by relaxing the 3D fields in the Atlantic zone (Fig. 1) to average vertical profiles which, for dissolved oxygen, phosphate, nitrate and silicate, refer to Salon et al. (2019), while ALKis based on what was described in Cossarini et al. (2015). These profiles do not consider a seasonal cycle or a future temporal evolution, with DIC as the only exception, which is prescribed from a global ocean-climate simulation under RCP8.5 emission scenario performed within the framework of the CMIP5 project (Coupled Model Intercomparison Project Phase 5; Taylor et al., 2012) and based on the CMCC-CESM modeling system (CMCC-Coupled Earth System Model; Vichi et al., 2011). The reasons for these choices rely on: (i) anomalous values observed in N:P ratio under the RCP8.5 emission scenario, (ii) negligible variation, under emission scenario RCP8.5, of ALK along the 21st century, (iii) lack of a consistent RCP4.5 scenario, (iv) the possibility, using the same conditions at the Atlantic boundary, to test the impacts of the different atmospheric and ocean forcings. Riverine inputs of phosphate, nitrate, dissolved oxygen, ALK and DIC are based on the PERSEUS FP7-287600 project dataset (Policy-oriented marine environmental research in the Southern European seas; Van Apeldoorn and Bouwman, 2014) and, also in this case, do not include temporal evolution in the future scenarios.

As observed in previous works (e.g. Richon et al., 2019), a transient scenario for the evolution of the atmospheric deposition of nitrogen and phosphorus over the Mediterranean Sea is presently not available. Following Di Biagio et al. (2019) and Reale et al. (2020a), the atmospheric deposition of phosphate and nitrate is parametrized as a mass flux at the surface and is set for the entire basin equal to 4780 Mmol year$^{-1}$ for phosphate and 81275 Mmol year$^{-1}$ for nitrate. Additional boundary conditions consider the sequestration of inorganic compounds in the marine sediment at the seabed.

The Representative Concentration Pathway (RCP) 4.5 and 8.5 emission scenarios (Moss et al., 2010) were used to force the coupled physical-biogeochemical MFS16-OGSTM-BFM system. RCP4.5 represents an intermediate scenario in which $CO_2$ emissions peak around 2040 (causing the maximum increase in $CO_2$ concentration), and then decline (with a resulting $CO_2$ concentration plateau) while the RCP8.5 represents the worst-case scenario, in which $CO_2$ emissions (eventually driven by feedback effects such as the release of greenhouse gasses from the permafrost) will continue to increase throughout the 21st century, and the $pCO_2$ concentration will rise to more than 1200 ppm at the end of the 21st century (IPCC, 2014). Recently some Authors have begun to consider the RCP8.5 scenario as "implausible", being based, for example, on a large use of coal, larger than its effective availability at the end of 21st century (e.g. Hausfather and Peters, 2020). On the other hand, it is still widely used to assess in the Mediterranean region the potential risks (also in the marine ecosystems) emerging in an extreme warm world climate (5 $^{o}$C) with respect to the pre-industrial era (IPCC, 2014). Because of that the projections under this emission scenario are still discussed here.

The initial conditions for the dissolved oxygen, nutrient, silicate and carbonate system variables are based on Medar-Medatlas dataset (Mediterranean Data Archeology and Rescue-Mediterranean Atlas), as described in Cossarini et al. (2015) and Salon et al. (2019).

Finally, all the simulations discussed in the next sections, use as initial conditions the resulting final fields from a run that started in January, 1st 2005 following a spin-up of 100 years made with a loop over the 2005–2014 period for the physical forcing, the river nutrient discharge and atmospheric forcing (nutrient deposition and $CO_2$ air value).

**2.4. Simulations protocol and set-up**

266

Long-term simulations can be affected by drifts in state variables due to the imbalance among boundary conditions, transport processes and internal element cycle formulations of the biogeochemical model. Therefore, a specific simulation protocol, based on the use of a control/scenario pair of simulation, has been implemented in order to disentangle the climate change signal from spurious signals (Solidoro et al., 2022). The protocol consists of a control simulation (CTRL) of 95 years and two 95-year biogeochemical scenario simulations, RCP4.5 and RCP8.5 (Fig.S1). All the simulations adopt as initial conditions the resulting final fields from the spin-up simulation (section 2.3). The CTRL is performed by repeating the 2005–2014 physical forcing and river discharge over the remaining 2015-2100 period (Fig. S1). The difference between each biogeochemical scenario and the CTRL provides the future evolution of a biogeochemical variable due to climate forcing.

Under each specific emission scenario and in the CTRL, our simulation protocol computes the time series of the mean annual 3D fields of the following variables: dissolved nutrients and oxygen, chl-*a*, net primary production, phytoplankton respiration, organic matter, plankton and bacterial biomass, POC, DIC and pH.

First, the annual 3D fields are vertically averaged over two separate key vertical levels: the surface zone and the intermediate zone. The first one spans the upper 100 m of the water column, which represents the location of MAW and the euphotic layer of the basin where most biological activities are concentrated. The second one covers the 200-600 m level, which includes the location of LIW. Only for the net primary production and phytoplankton respiration, a vertical integral over the 0-200 m layer is considered (Lazzari et al., 2012).

Second, the temporal evolution of the unbiased scenario starting from the present state, $U(k)_{SCEN}$ (with $k = 2005, ..., 2099$), is defined as:

$$U(k)_{SCEN} = X'_{SCEN} + X(k)_{SCEN} - X(k)_{CTRL} \qquad (1)$$

where $X'_{SCEN}$ is the average of $X(k)_{SCEN}$ over the 2005-2020 period (hereafter the PRESENT, Fig.S1), and $X(k)_{SCEN}$ and $X(k)_{CTRL}$ are the yearly average in the scenario and CTRL simulations, respectively. We introduce the concept of "unbiased scenario" because equation (1) removes the effect of potential model drifts due to unbalanced boundary conditions and model errors. The time series of CTRL are filtered with a linear regression to keep the long-term drift and remove spurious variability. The period 2005-2020 has been chosen as reference (also in the forthcoming validation) due to: (i) the availability, after year 2000, of more advanced satellite and assimilated datasets to evaluate the biogeochemistry of the basin, (ii) to avoid the overlapping between historical and scenario part of the simulations (with the latter starting in 2005). It is important to stress here that the choice of the period should not significantly affect the results of the study as the observed differences during this period between the two scenarios for temperature, salinity and current speed fields have been found to be not statistically significant over most of the basin (not shown).

Finally, the temporal evolution of the climate change signal (CCS) with respect to the present is given by:

$$CCS(k)_{SCEN} = U(k)_{SCEN} - U_{SCEN-PRESENT} \qquad (2)$$

where $U_{SCEN-PRESENT}$ is the average of $U(k)_{SCEN}$ in the PRESENT. Hereafter, if not differently specified, all the shown time series will be represented by $CCS_{SCEN}$.

305

Horizontal spatial averages are computed considering the sub-basins defined in Fig. 1, the whole Mediterranean basin scale, and two macro-areas: the Western Mediterranean (WMED which includes ALB, SWM, NWM, TYR) and the Eastern Mediterranean (EMED which includes ION and LEV). The Adriatic and Aegean Seas are usually not considered part of the Eastern Mediterranean due to the importance of local forcing, such as riverine loads, in shaping the variability of the biogeochemical dynamics in those two sub-basins. Because of that, following the approach already adopted in previous works (Lazzari et al., 2012; 2016; Di Biagio et al., 2019; Reale et al., 2020 a,b) they are not considered in the spatial averages related to WMED and EMED.

Temporal averages of the climate change signals are computed over two 20-year periods: 2040-2059, hereafter referred to as "MID-FUTURE" and 2080-2099, hereafter referred to as "FAR-FUTURE" (Fig.S1). The relative climate change signals (in %, except for pH which will be measured in units of pH) in the MID-FUTURE or FAR-FUTURE periods with respect to the PRESENT are computed as:

$$U_{MID-FUTURE}=100*(U_{SCEN-MID-FUTURE}-U_{SCEN-PRESENT})/U_{SCEN-PRESENT} \quad (3)$$

$$U_{FAR-FUTURE}=100*(U_{SCEN-FAR-FUTURE}-U_{SCEN-PRESENT})/U_{SCEN-PRESENT} \quad (4)$$

where $U_{SCEN-MID-FUTURE}$, $U_{SCEN-FAR-FUTURE}$ and $U_{SCEN-PRESENT}$ are the averages of $U(k)_{SCEN}$ for the MID-FUTURE, FAR-FUTURE and PRESENT periods, respectively. Hereafter, if not differently specified, all the percentages shown in the maps are represented by $U_{MID-FUTURE}$ and $U_{FAR-FUTURE}$. The statistical significance of the relative climate change signals in each point of the basin is assessed by means of Mann-Whitney test with p<0.05.

**3. Results**

**3.1 Evaluation of the MFS16-OGSTM-BFM control simulation for the present climate**

MFS16 modelling system performances under present climate conditions were previously analyzed (Lovato et al., 2013; Galli et al., 2017), showing that the main spatio-temporal characteristics of the Mediterranean Sea physical properties reliably compared against ocean reanalysis datasets. Moreover, the physical reanalysis dataset produced by MFS16 within the Copernicus Marine Environmental Marine Service (CMEMS, Simoncelli et al., 2019) has already been coupled to the transport-reaction model OGSTM-BFM to carry out a reanalysis for the Mediterranean Sea biogeochemistry (Teruzzi et al., 2019). The latter is a biogeochemical dataset covering the 1999-2015 period at 1/16º resolution, which was already used for validating different biogeochemical simulations in the Mediterranean Sea, such as those based on MEDMIT12-BFM (Mediterranean MIT General circulation Model-BFM at 1/12º; Di Biagio et al., 2019) and RegCM-ES (Regional Climate Model-Earth System; Reale et al., 2020a) modelling systems. This dataset has been recently upgraded, refining the resolution to 1/24º and extending the period to 2019 (Teruzzi et al., 2021; Cossarini et al., 2021).

To date, no future climate biogeochemical projection of the Mediterranean Sea has been performed through this offline coupling.

Figure 2 a,b shows the surface average chl-*a* concentrations (upper 10 m) from the CTRL run compared with a
climatology based on satellite data available from CMEMS which covers the period 1999-2015 (Colella et al., 2016). The
model correctly reproduces the areas in the Mediterranean region characterized by relatively high values of chl-*a*: the
Alboran Sea, the Gulf of Lion, the coastal areas of the Adriatic Sea, and the Strait of Sicily. Moreover, the CTRL
simulation captures the west-east trophic gradient of chl-*a*, whose existence has been pointed out in previous works
(D'Ortenzio and Ribera d'Alcala, 2003; Lazzari et al., 2012; Colella et al., 2016; Richon et al., 2019; Di Biagio et al.
2019; Reale et al., 2020a). On the other hand,  a general underestimation of approximately 50% of the chl-*a* signal
throughout the basin and in the coastal areas is observed, probably associated with insufficient river load (Richon et al.,
2019; Reale et al., 2020a) and with the tendency of satellite chl-*a* measures to be systematically overestimated in the
coastal areas with respect to "in situ" observations due to the presence of particulate suspended matter in the water column
(Claustre et al., 2002; Morel et Gentili, 2009).
Figure 2 also shows the average vertical profiles, computed for the entire, Western and Eastern Mediterranean basins, of
chl-*a* (c), $PO_4$ (d), $NO_3$ (e), dissolved oxygen (f), DIC (g), pH (h) and ALK (i) in the CTRL compared with the recent
CMEMS reanalysis (only for chl-*a* and pH, Cossarini et al., 2021) and EMODnet datasets (European Marine Observation
and Data Network; Buga et al., 2018). In spite of the tendency to overestimate the chl-*a* values, the model captures the
DCM location, the west-east trophic gradient in the basin, and also the nutricline depths deepening between Western and
Eastern basin and the low nutrient surface concentrations. Mean simulated values in the first 0-200 m are quite realistic
for almost all the biogeochemical tracers and properties, with correlation values between observations and modelled data
greater than 0.93. At the same time, the CTRL overestimates the $PO_4$ concentration between 100 and 300 m of about 50%,
and the dissolved oxygen concentration of about 15% below 200 m and underestimates, below 200 m, the $NO_3$
concentration of about 20% and the pH of about 1 % between 100 and 300 m. It is worthwhile to point out that the limited
spatial resolution of the observations below 200 m could impact the robustness of our comparison. In general, the biases
in the initial conditions are originated by the spin-up simulation that allows to remove the largest part of model drifts. As
explained in section 2.4, these biases, which are still present in both the CTRL and scenario simulations, do not affect the
calculation of the climate change signals, and are generally lower than the changes observed in the scenarios at the end
of the century.
To summarize, although the model shows some deficiencies in simulating the vertical distribution of some
biogeochemical tracers and properties, the main features of the system are reliably simulated and thus, MFS16-OGSTM-
BFM is robust enough to be used to investigate the evolution of the Mediterranean biogeochemistry under different
emission scenarios.

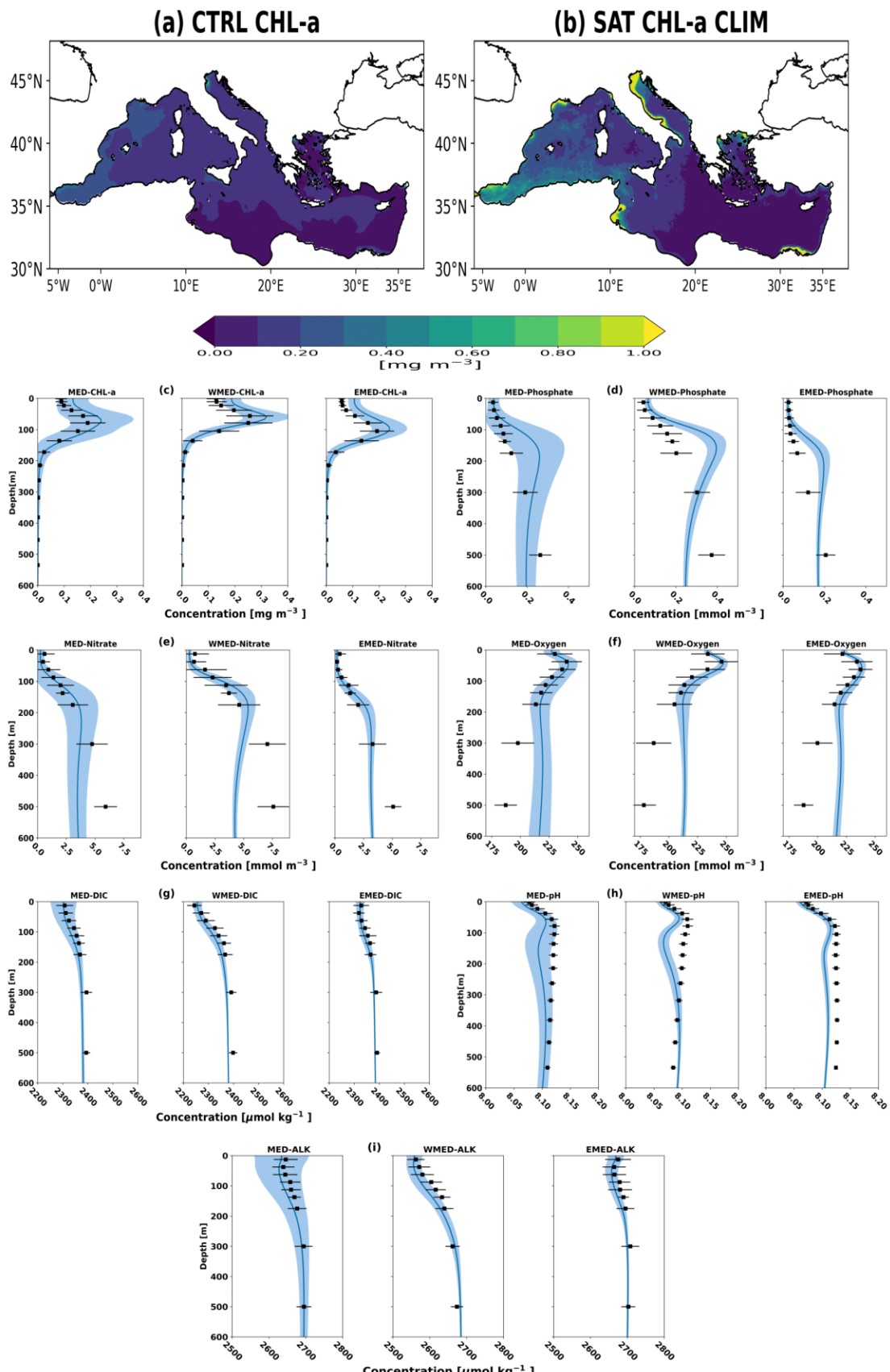



**Fig.2 Average chl-a in the first 10m in CTRL (a) for the period 2005-2020 and CMEMS-SAT (b) together with CTRL average**
**vertical profiles (blue lines) for the period 2005-2020 of chl-a (c, mg m⁻³), PO₄ (d, mmol m⁻³), NO₃ (e, mmol m⁻³), Dissolved**

oxygen (f, mmol m$^{-3}$), DIC (g, μmol kg$^{-1}$), pH (h) and ALK (i, μmol kg$^{-1}$). The averaged profiles are computed for the entire (MED), Western (WMED) and Eastern (EMED) Mediterranean Sea. The light blue areas represent the spatial standard deviation of the monthly model data. The model data are compared with CMEMS reanalysis (chl-a and pH; Colella et al., 2016: Teruzzi et al., 2021) and observations provided by EMODnet (PO$_4$, NO$_3$, Dissolved oxygen, DIC, ALK; Buga et al., 2018): annual mean (black squares) and related standard deviations (black bars). Depth is measured in meters**.

**3.2 Evolution of the thermohaline properties and circulation of the Mediterranean Sea in the 21st century**

Mean temperature and salinity evolution between 0-100 m and 200-600 m in the 2005-2099 period under the RCP4.5 and RCP8.5 scenarios in the whole Mediterranean Sea and in the Western and Eastern basins are shown in Fig. 3. As for the biogeochemical variables, these depths have been chosen as they are representative of the location of MAW and LIW, respectively.

A warming of the surface and intermediate layers is observed at the basin scale and in both the Western and Eastern basins, whose magnitude (approximately 1.5 °C in the RCP4.5 and 3°C in the RCP8.5 scenario) agrees with what has already been observed in recent modelling studies based on single/multimodel ensembles (e.g., Adloff et al., 2015; Soto-Navarro et al., 2020).

Similar to the seawater temperature, the variation in salinity is strongly dependent on the emission scenario with more intense anomalies, both negative and positive, under RCP8.5 conditions (as observed in previous modelling studies such as Adloff et al., 2015 and Soto-Navarro et al., 2020). For example, salinity in the surface layer at basin scale and in the Eastern basin is characterized by a decrease between 2020 and 2050 followed by a constant increase (stronger under RCP8.5 scenario) until the end of the 21st century. Conversely, after 2050, the Western basin shows a freshening of the surface layer with respect to the beginning of the century, in agreement with what was already observed by Soto-Navarro et al. (2020). An increase in salinity also occurs in both scenarios in the intermediate layer both at the basin scale and in the two main sub-basins.

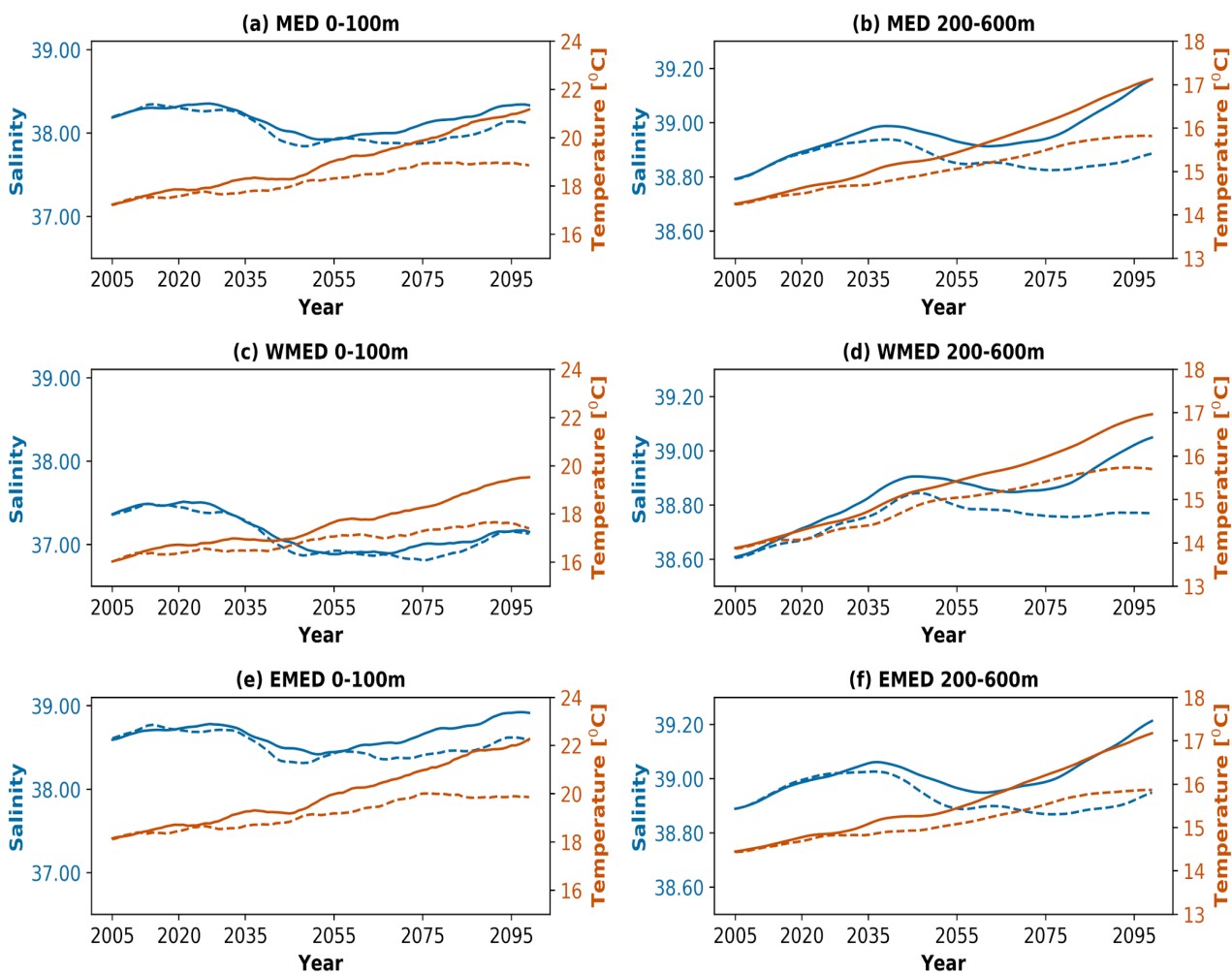

**Fig.3 - Yearly time series for the period 2005-2099 of Salinity (blue) and Temperature (dark orange, in ºC) under the emission scenarios RCP8.5 (solid line) and RCP4.5 (dashed line) in the Mediterranean Sea (MED, a-b), Western Mediterranean (WMED, c-d) and Eastern Mediterranean (EMED, e-f) for the layers 0-100 m (left column) and 200-600 m (right column). The yearly time series have been smoothed using 10-years running mean.**

The spatial distribution of temperature variations in the surface layer (Fig. S2) shows a comparable and mostly statistically significant on basin-scale warming in RCP4.5 and RCP8.5 in the MID-FUTURE (the differences between the projected changes are lower than 2%), while, in the FAR-FUTURE, the projected changes in the RCP4.5 are approximately the 50% lower with respect those observed in RCP8.5 (8-12% and 17-20% respectively), with the North Western Mediterranean, Tyrrhenian, Adriatic, Ionian, Aegean Sea and Levantine, being the most affected sub-basins. Local relative maxima are observed in both scenarios, in the Gulf of Lion, in the relatively shallow and in the coastal areas of the Adriatic Sea and in the area of the Rhodes Gyre (Fig.S2 i,j).

A general freshening of the upper layers and saltening of the intermediate layers over most of the Mediterranean basin is observed during the MID-FUTURE period (Fig. S3). The projected changes are statistically significant over most of the basin with the only exception, in both scenarios, of the upper layer of the Adriatic Sea and Northern Ionian Sea and the intermediate layers of the Southern Ionian and Levantine Basin/Southern Adriatic and Northern Ionian Sea in the RCP4.5/RCP8.5 scenario as consequence, probably, of the river input in the Adriatic Sea and mid-Ionian Jet dynamics.

The latter has been recognized, in fact, as an important driver for the salinity for the upper and intermediate layers of the
Adriatic and Ionian Sea (e.g. Gacic et al., 2010). In the FAR-FUTURE, the freshening of the surface is still present at the
basin scale in the RCP4.5 scenario (although it is reduced with respect to the MID-FUTURE) and in the Western basin
in the RCP8.5 scenario. Moreover, an increase in salinity is observed in the Adriatic Sea and in the Eastern basin under
RCP8.5. The projected changes in the surface salinity in the Adriatic Sea and Northern Ionian Sea under RCP4.5 are also
not significant.

The decrease in salinity in the 21st century in the Western basin is driven by the salinity values imposed in the Atlantic
buffer zone (Lovato et al., 2013), while the saltening of the Eastern basin, under RCP8.5 scenario, is linked to the
decreasing freshwater discharge in the area (e.g., Gualdi et al., 2013; Soto-Navarro et al., 2020). In the intermediate layer,
the situation is reversed: while in RCP8.5 the entire basin experiences an increase in the salinity associated with the
increase in salinity in the surface water of the Eastern basin, in RCP4.5 the Eastern basin experiences a slight decrease in
salinity associated again with the freshening of surface water. In fact, at the surface, both signals are transported by
vertical mixing to the intermediate layers of the Eastern basin influencing the salinity of the newly formed LIW.

Figure 4 shows the temporal evolution of the Mediterranean thermohaline circulation during the 21st century using the
zonal overturning stream function (or ZOF; Myers and Haines, 2002; Somot et al., 2006). The ZOF has been computed
by the meridional integration from south to north and from the bottom to the top of the water column of the zonal velocity
(see Adloff et al., 2015). The domain of the integration is the same as shown in Figure 1 with the exclusion of the Atlantic
area outside the Strait of Gibraltar. The thermohaline circulation of the basin in the PRESENT is composed of two cells,
similar to the outcomes of the historical reference experiments described in Adloff et al. (2015) and Waldman et al.
(2018). The first cell extends from the surface to 800 m, with a clockwise circulation associated with MAW moving
eastwards and LIW moving westwards. The second cell is located between 500 and 2500 m in the Eastern Mediterranean
with a counterclockwise circulation associated with the Eastern Mediterranean Deep Water (EMDW) moving eastwards
and LIW moving westwards.

Under the two scenarios, during the MID-FUTURE period, there is an evident weakening of both cells and a reduction
of the thickness of the upper layer cell and the Eastern basin cell (less than -0.1 Sv), which splits into two sub-cells. By
the end of the century both cells show a similar behavior, whereas in the RCP4.5 scenario, the Eastern cell is slightly
more intense. The weakening of the zonal overturning stream function is similar to previous findings of Somot et al.
(2006) and Adloff et al. (2015). As the Mediterranean thermohaline circulation is driven by both deep and intermediate
water formation processes, the overall weakening of both cells is a direct consequence of the increase in the vertical
stratification of the water column. In fact, the evolution of the winter maximum mixed layer depth in key convective areas
of the Mediterranean Sea, such as the Gulf of Lion, Southern Adriatic, Aegean Sea and Levantine basin (Fig. S4), shows
a progressive decrease in the intensity of the open ocean convection after 2030. Only for the Aegean Sea, the changes in
the winter mixed layer maximum depth are less marked, with the occurrence of some maxima around 2080 (in RCP8.5)
or after 2090 (in RCP4.5), which could correspond to a future tendency of the thermohaline circulation of the Eastern
basin to produce EMT (Eastern Mediterranean Transient)-like events (Adloff et al., 2015).

The projected overall weakening of the Mediterranean thermohaline circulation leads to a reduction in the exchanges of
biogeochemical properties between the Western and Eastern basins through the Strait of Sicily at both the surface and
intermediate levels (Fig. S5) and to a reduced ventilation of intermediate/deep waters (Adloff et al., 2015).

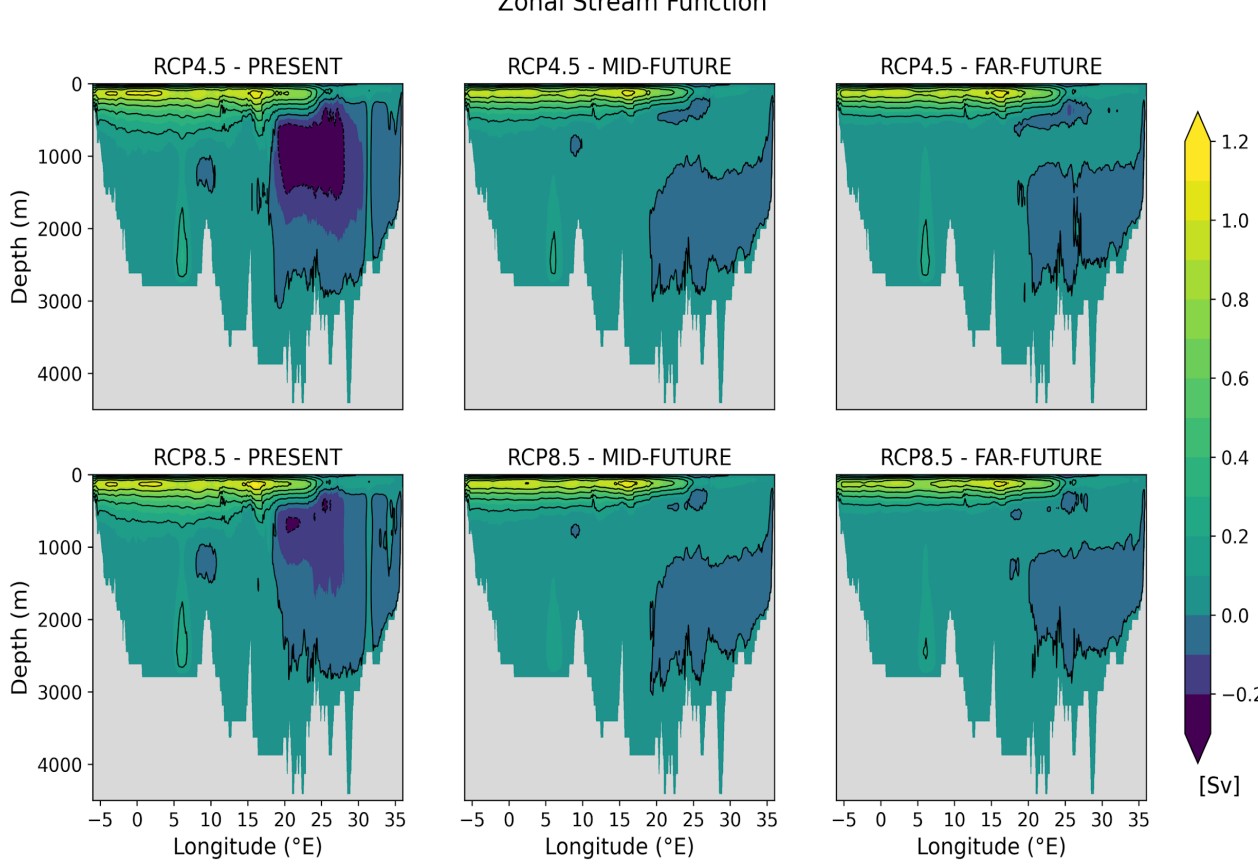

**Fig. 4 - Mediterranean Sea zonal stream function annual mean (in Sv) averaged over the PRESENT (2005-2020), MID-**
**FUTURE (2040-2059) and FAR-FUTURE (2080-2099) periods under RCP4.5 and RCP8.5 scenarios.**

**3.3 Spatial and temporal evolution of nutrients, dissolved oxygen and chl-*a* concentrations**

Figures 5 and 6 show the spatial distribution of the magnitude and signs of the changes that will affect the dissolved
nutrient concentrations during the 21st century. In the FAR-FUTURE, the decreases in $PO_4$ and $NO_3$ concentrations in
the 0-100 m layer are almost half in RCP4.5 (approximately 7% and 13% for $PO_4$ and $NO_3$, respectively) with respect to
those observed in the RCP8.5 (approximately 13% and 20% for $PO_4$ and $NO_3$, respectively) and are particularly marked
and statistically significant in the Levantine basin, in the Aegean Sea, in the Central/Southern Adriatic Sea and Northern
Ionian Sea. Again, statistically significant relative local maxima (in absolute value) are observed in both scenarios in the
area of the Gulf of Lion, Southern Adriatic, Northern Ionian and Rhodes Gyre. Moreover, there are clear spatial gradients
affecting the statistical significance of the projected changes. For example, the projected changes in nutrient concentration
in the Northern Adriatic Sea and in many other coastal areas, influenced by river dynamics, are not significant, contrary
to what is observed in the open ocean areas of the same sub-basin. Here, the projected decrease is associated with the
reduced vertical mixing in the water column and reduced inflow of nutrients through the Otranto Strait (Fig. S6). Finally,
the two scenarios show some significant changes in the dissolved nutrient concentrations at local scale in the Alboran Sea
and in the Southern Ionian associated with changes in the intensity of mesoscale circulation (eddies) of both areas and in
the intensity and spatial structure of the mid-Ionian jet (not shown).

In contrast to the general decreasing nutrient content of the upper layer, the intermediate layer in both scenarios shows a
strong (milder) increase in nutrient concentration in the Southern Aegean Sea (Levantine basin, North Western
Mediterranean and Alboran Sea) in the 21st century driven by the reduced vertical mixing, which tends to increase the
nutrient content of the intermediate layers. The Tyrrhenian, Ionian and Southern Adriatic Seas are, in turn, characterized
by a permanent negative anomaly. In the first two areas, the anomaly can be associated with the decrease in the westward
transport of nutrients in the intermediate layers through the Strait of Sicily (consequences of the weakening of the zonal
stream function discussed in Section 3.2, Fig. S5), while in the Adriatic Sea, the projected changes are driven by the
increase in the nutrient export in the intermediate layer through the Otranto Strait (Fig. S6). In the North Western
Mediterranean, the observed positive anomalies become weaker and even negative in the FAR-FUTURE under the
RCP4.5 emission scenario, likely due to some convective events that take place between 2080 and 2100, as shown in Fig.
S4. Comparing the projected changes at the surface in the FAR-FUTURE it is observed that while under RCP4.5 in most
of the Western Mediterranean and the Ionian Sea they are not statistically significant, under RCP8.5 emission scenario
the statistical significance that was initially limited to Adriatic, Aegean Sea and Levantine basin, now it also involves the
Ionian and Tyrrhenian Sea.

PO$_4$

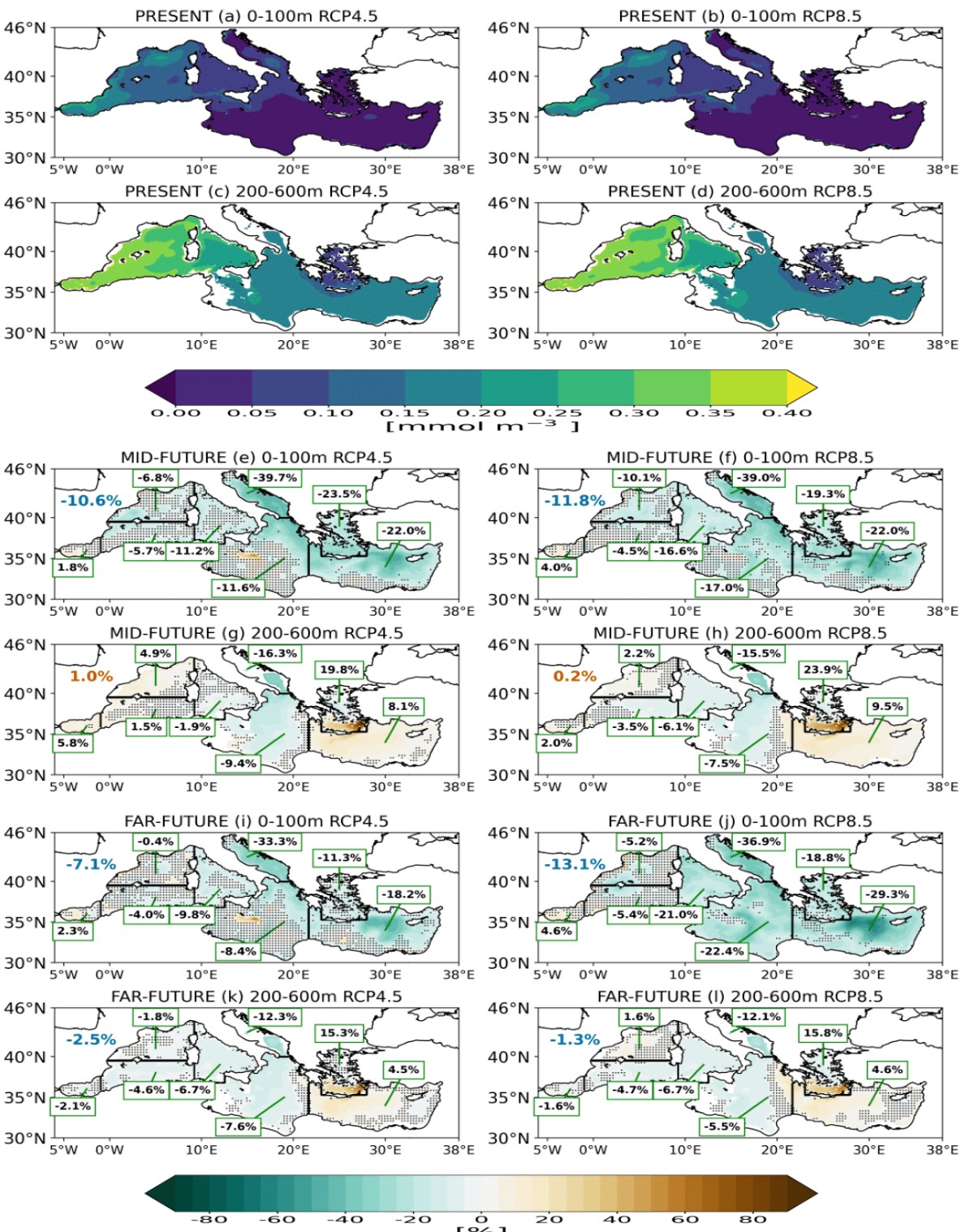


**Fig. 5 - Phosphate concentration (in mmol m-3) in the layers 0-100 m and 200-600 m in the PRESENT (2005-2020, a,b,c and**
**d), and relative climate change signal (with respect to the PRESENT) in the MID-FUTURE (2040-2059, e,f,g and h) and FAR-**
**FUTURE (2080-2099, i,j,k and l) in the RCP4.5 (left column) and RCP8.5 (right column) emission scenario. The Mediterranean**
**average relative climate change signal in each period (with respect to the PRESENT) is displayed by the top-left colored value**
**(blue or dark orange when negative or positive). Values in the green boxes are the average relative climate change signal in**
**each period and in each sub-basin shown in Figure 1. Domain grid points where the relative climate change signals are not**
**statistically significant according to a Mann-Whitney test with p<0.05 are marked by a dot.**

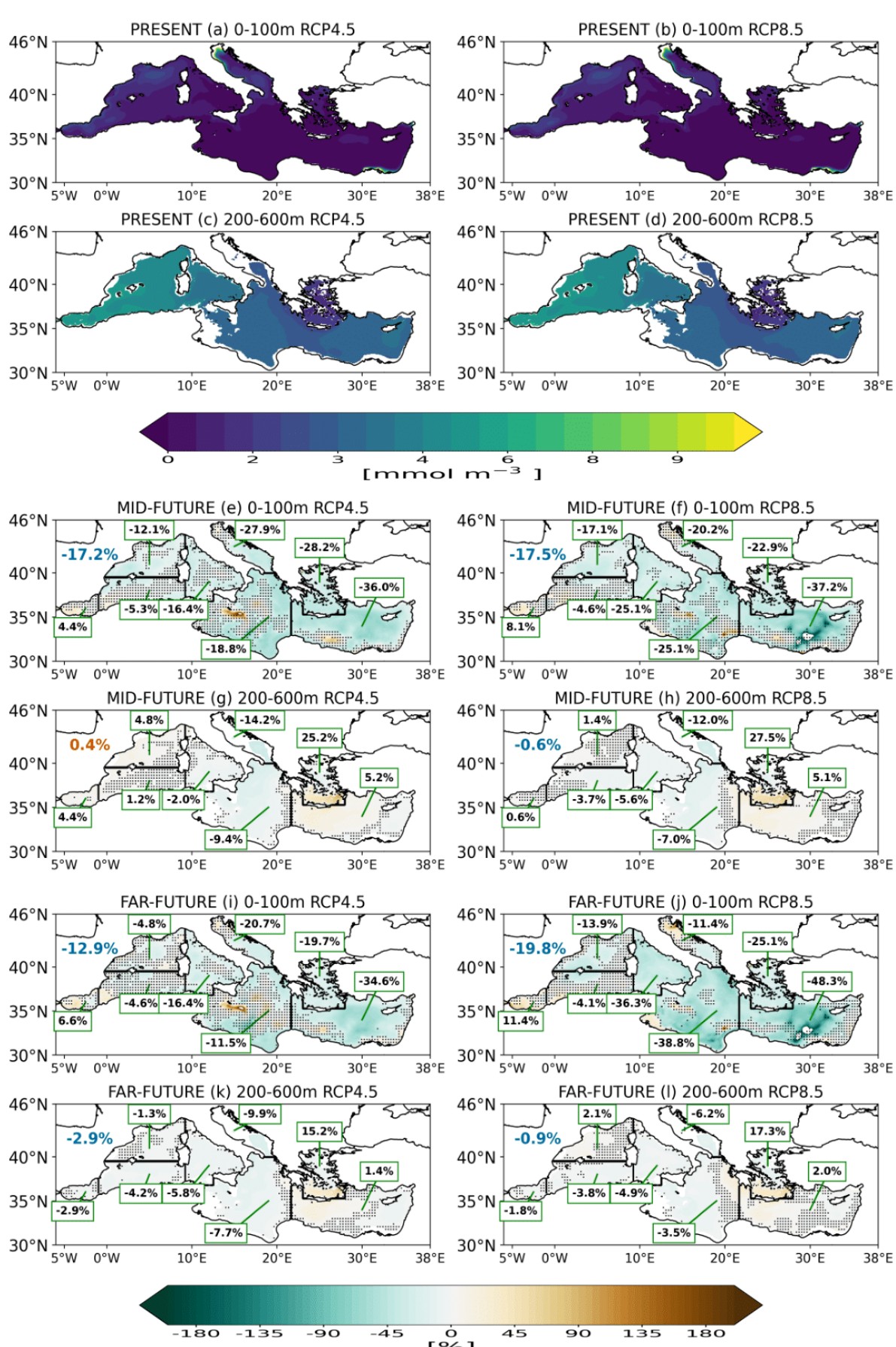

NO$_3$



**Fig.6 - as Fig.5 but for Nitrate (in mmol m$^{-3}$)**





The temporal evolution of the mean concentrations of PO$_4$ and NO$_3$ in the RCP4.5 and RCP8.5 simulations between 0-
100 m and 200-600 m in the Mediterranean Sea and its Western and Eastern basins for the 2005-2099 period is shown in
Fig. 7. In the RCP8.5 scenario, PO$_4$ and NO$_3$ concentrations within the euphotic layer of both sub-basins are substantially
stable for the first 30 simulated years, while a marked decline occurs after 2030-2035, with values of 0.01 and 0.1 mmol
m$^{-3}$ (compared to the beginning of the century) respectively, which is followed by a steady evolution of the concentration
values until the end of the century. The same behaviour is observed in RCP4.5, except for a recovery that takes place at
the end of the century in correspondence to an increase in the nutrient inflow into the Alboran Sea (Fig. S7). The observed
decline is timely in phase with the weakening of the zonal stream function discussed in Fig. 4, further pointing out the
importance of the vertical mixing in driving the temporal variability of nutrients in the euphotic layer. From this point of
view, some relative maxima of both nutrient concentrations in the Western and Eastern basins are observed for RCP4.5
in the 2015-2040 period (Fig. 5 c,d), associated with strong ocean convective events taking place in the Gulf of Lion and
Levantine basin (Fig. S4). Between 2055 and 2075, the peak in both nutrients' concentration, for RCP4.5, timely
corresponds to a peak in the inflow of nutrients into the Alboran Sea (Fig. S7). Additionally, in both the scenarios, the
intermediate layer of the Western basin, after 2035, experiences a negative tendency in the nutrient concentration (greater
than 0.01 mmol m$^{-3}$ for PO$_4$ and 0.1 mmol m$^{-3}$ for NO$_3$) related to a reduced westward transport of nutrients associated
with LIW (Fig.S5).

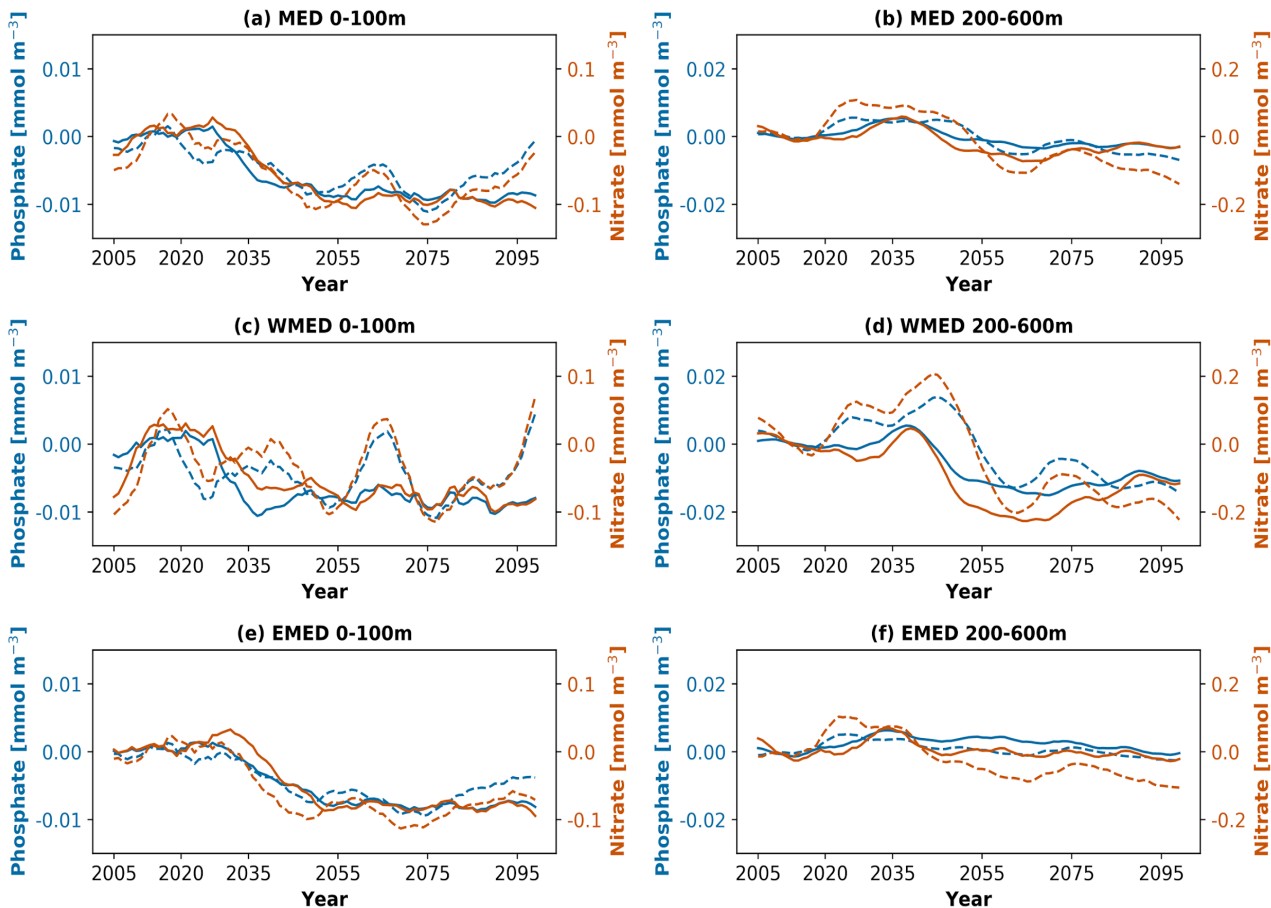


**Fig.7 - Yearly time series for the period 2005-2099 of Phosphate (blue, in mmol m$^{-3}$) and Nitrate (dark orange, in mmol m$^{-3}$)**
**anomalies for the emission scenario RCP8.5 (solid line) and RCP4.5 (dashed line) in the Mediterranean Sea (MED, a-b),**
**Western Mediterranean (WMED, c-d) and Eastern Mediterranean (EMED, e-f) for the layer 0-100 m (left column) and 200-**
**600 m (right column). The yearly time series have been smoothed using 10-years running mean.**

The temporal evolution of chl-*a* in the two scenarios is similar to what was observed in the case of dissolved nutrients,
with a high interannual variability, a decrease after 2030-2035 of approximately 0.03 mg m$^{-3}$ and a stable signal until the
end of the century in the RCP8.5 scenario while, in the case of the RCP4.5 a recovery towards the observed PRESENT
values is simulated at the end of 21st century (Fig.8). In the Eastern Mediterranean the decrease is of the same magnitude
as that observed at the basin scale, while in the Western basin the chl-*a* signal appears substantially stable with respect to
the present.

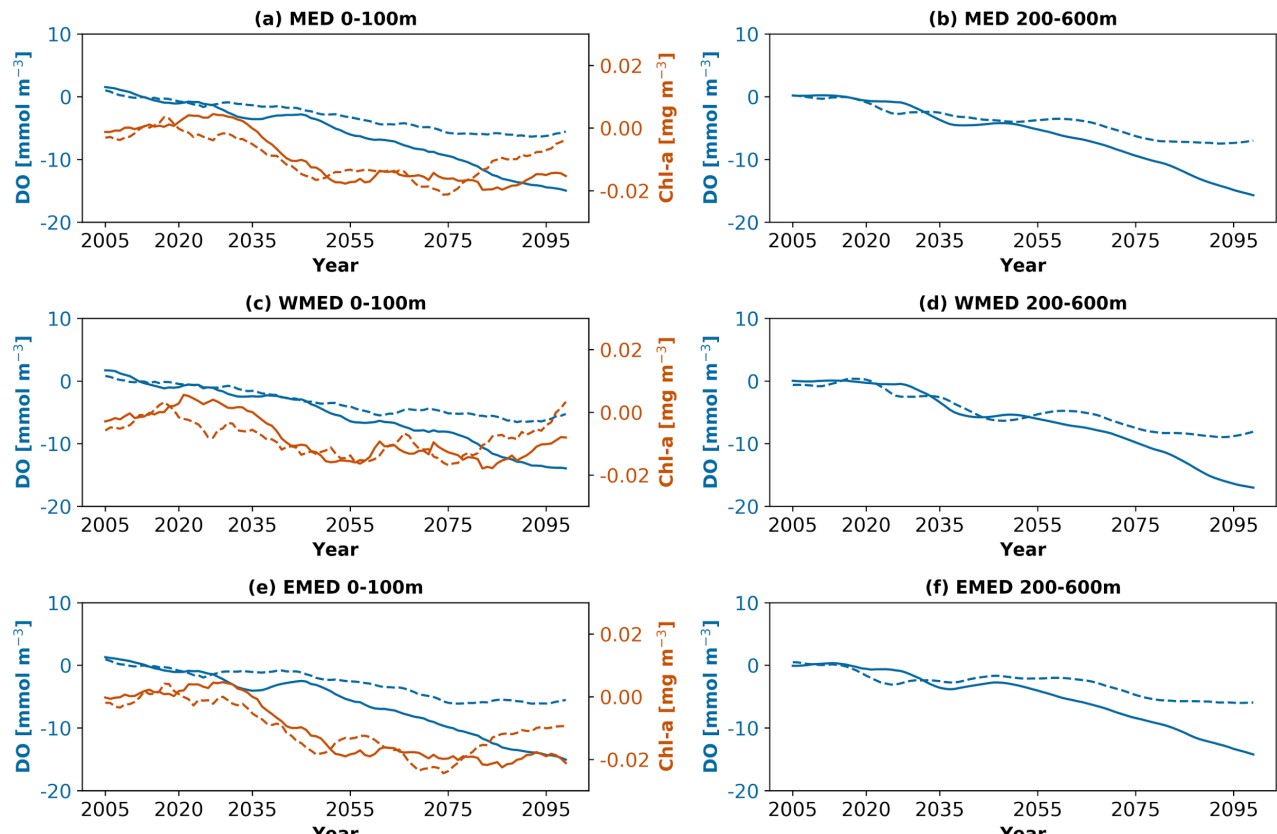


**Fig. 8 as Fig.5 but for Dissolved Oxygen (blue, in mmol m⁻³) and Chl-*a* (dark orange, in mg m⁻³)**

During the 21st century, a continuous decrease in the oxygen concentration is projected in both scenarios in the
Mediterranean Sea (Fig. 8). The simulated reduction of the oxygen values is slower in the RCP4.5 with respect to RCP8.5.
For example, under the RCP8.5 emission scenario, the concentration of the dissolved oxygen in the upper layer decreases
by approximately 15 mmol m⁻³, which is three times the value observed in the RCP4.5 scenario (Fig. 8). The decrease in
dissolved oxygen is rather uniform and almost statistically significant everywhere in both the horizontal and vertical
directions in all the sub-basins, with values that are half in RCP4.5 (in percentages) with respect to those observed under
RCP8.5 (Fig. S8). For example, the decrease in the oxygen concentrations in the Levantine basin, in the FAR-FUTURE,
is approximately equal to 3% under the RCP4.5 emission scenario and 6% under the RCP8.5 emission scenario. In the
North Western Mediterranean, these values are approximately 3% and 7% respectively. The projected decreases in both
scenarios are usually lower in the Alboran Sea and South Western Mediterranean with respect to the rest of the basin, as
a consequence of the damping effect driven by the oxygen values imposed at the Atlantic boundary. In fact, the advection
of dissolved oxygen associated with AW partially limits the reduction in the oxygen solubility at the surface as a
consequence of the warming of the water column in the sub-basins near the Strait of Gibraltar, such as the Alboran Sea.

The uniform decrease in the oxygen surface concentration observed in Fig. S8 is spatially coherent (also from the
statistical point of view) with the increase in the temperature shown in Fig. S4, confirming the importance of temperature
in driving the solubility of the oxygen in the marine environment (Keeling et al., 2010; Shepherd et al., 2017). Moreover,
a decrease in the oxygen inflow (not shown) into the Alboran Sea and an overall increase in community respiration (see
the analysis related to the phytoplankton respiration in section 3.4) are found, which represent additional factors
explaining the projected changes. Western sub-basins, deep convection areas and shallow coastal zones of the Adriatic

Sea are the regions that show the highest decrease of oxygen in both surface and intermediate layer, with again the magnitude of the observed signal depending on the considered scenario (Fig. S8) and related to the reduction in vertical processes's intensity. The effect of the increased stratification on the oxygen vertical distribution is clearly shown in Figure 9. Under RCP8.5 (Fig.9 a,b), the progressive decline of oxygen concentration is timely corresponding to the progressive decrease in the maximum mixed layer depth (Fig. S4) and the weakening of the zonal stream function (Fig.4) discussed in Section 3.2. For example, in the North Western Mediterranean the correlation coefficient between the average dissolved oxygen concentration in the first 100 m and the maximum mixed layer depth has been found equal to 0.64 (statistically significant with p<0.05). On the other hand, under the RCP4.5 emission scenario, some events of deep transport of oxygen, that dumped the decline in the oxygen concentration, can be recognized towards the end of the 21st century in both Western and Eastern Mediterranean.

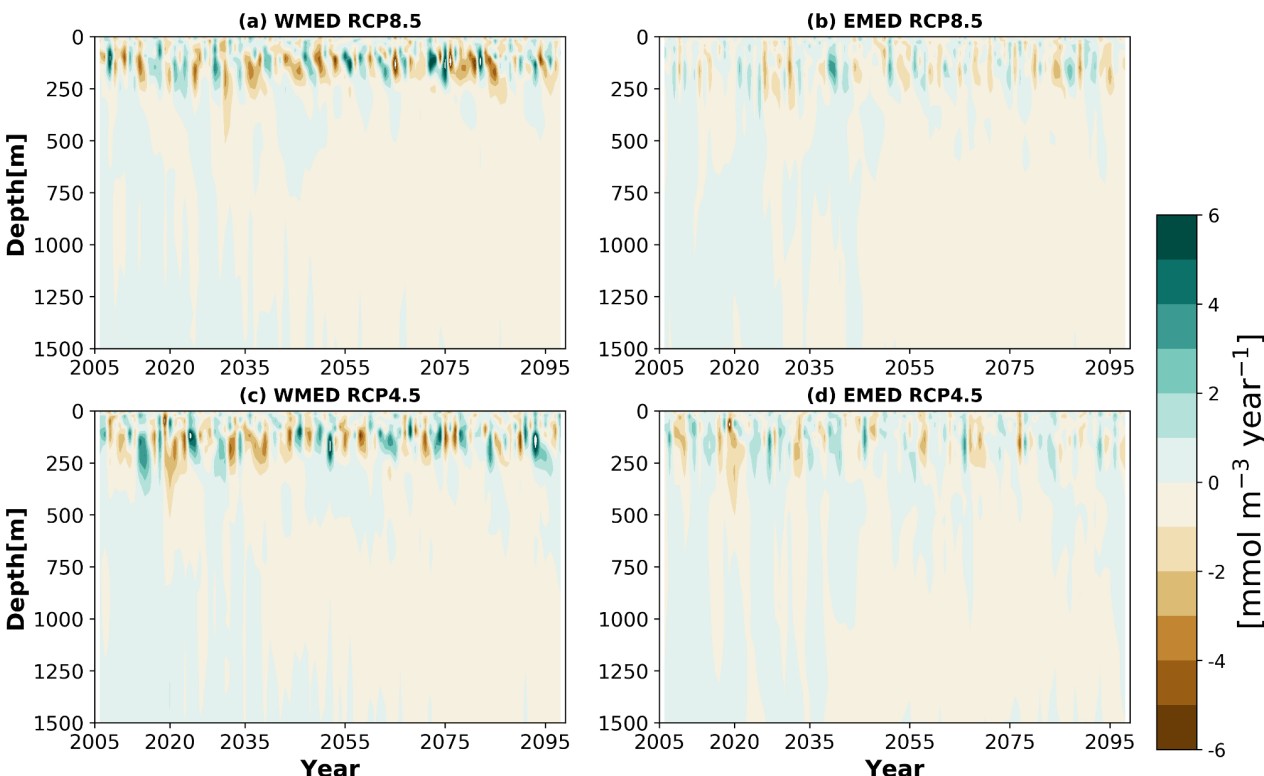

**Fig.9 Annual rate of change of Dissolved Oxygen (mmol m$^{-3}$ year$^{-1}$) in the Western (a,c) and Eastern (b,d) Mediterranean Sea in RCP8.5 (a,b) and RCP4.5 (c,d).**

**3.4 Spatial and temporal evolution of net primary production and living/non-living organic matter**

The warming of the water column and the increase in vertical stratification affect the metabolic rate of ecosystem processes including $CO_2$ fixation and community respiration. In fact, a basin-wide increase in net primary production (NPP) starting after 2035 and proceeding until the end of the simulations, is projected in both scenarios (Fig. 10). In the RCP4.5 scenario the NPP increase is greater than 10 gC m$^{-2}$ year$^{-1}$, which is a value that is more than half with respect to the values observed in the RCP8.5 simulation.

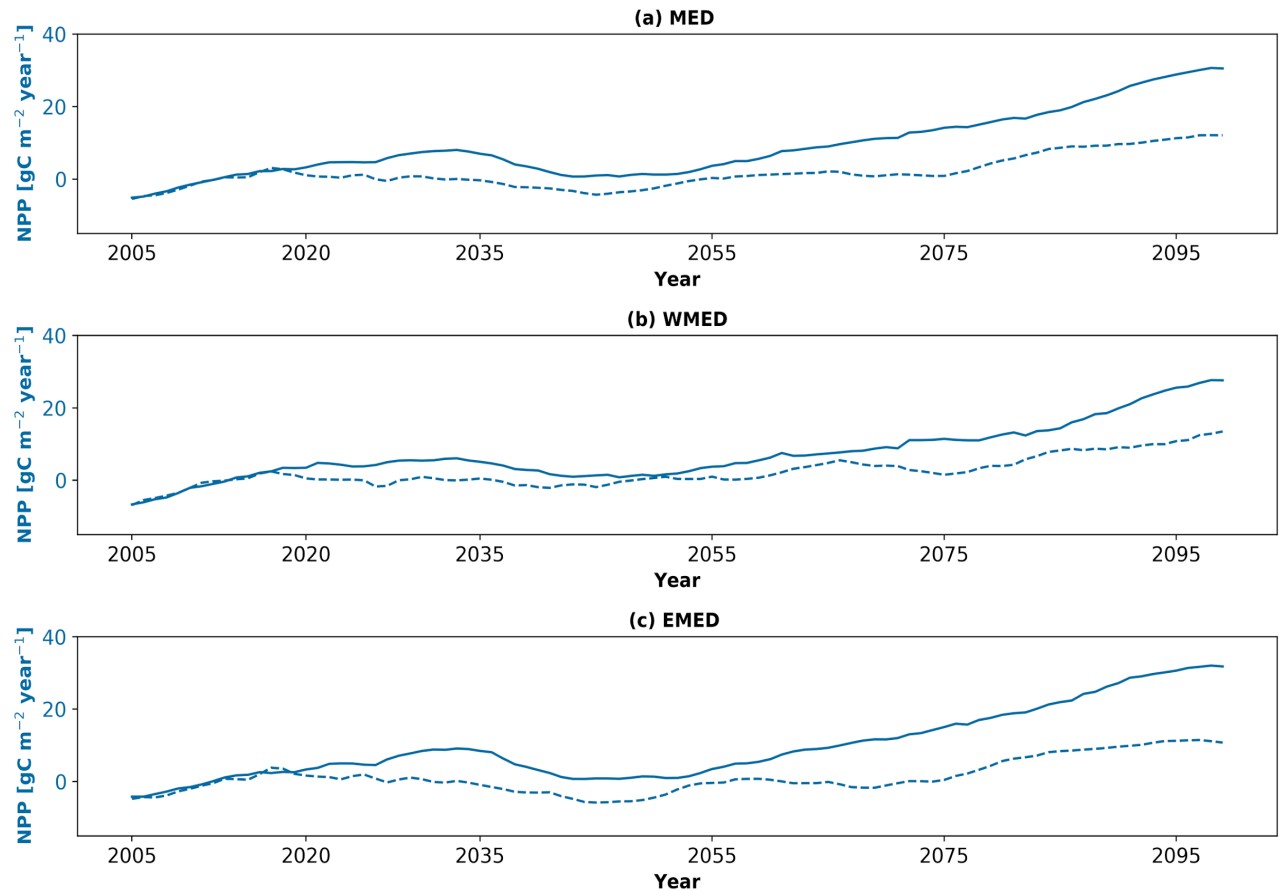

599

**Fig.10 - Yearly time series for the period 2005-2099 of Integrated net primary production (blue, in gC m⁻² year⁻¹) anomalies for**
**the emission scenario RCP8.5 (solid line) and RCP4.5 (dashed line) in the Mediterranean Sea (MED, a), Western**
**Mediterranean (WMED, b) and Eastern Mediterranean (c) for the first 200 m. The yearly time series have been smoothed**
**using 10-years running mean.**

The distribution of the sign of the NPP changes is not uniform across the basin and between the simulations (Fig.11). In
the MID-FUTURE, in both scenarios, the only areas that experience an increase (not statistically significant in all the
cases) in the NPP with respect to the beginning of the century are the North Western Mediterranean, the Tyrrhenian Sea,
the Northern Adriatic Sea, part of the Ionian Sea and of the Levantine basin. Conversely, the only statistically significant
projected changes are negative and are observed in the Central and Southern Adriatic Sea, part of the Northern Ionian
Sea and the Rhodes Gyre areas. The Aegean Sea shows a rather opposite behavior with a negative/positive anomaly in
RCP4.5/RCP8.5. In the FAR-FUTURE, corresponding to a more pronounced warming of the basin, the NPP increase is
quite uniform and statistically significant over most of the basin and is equal approximately equal to 7% in RCP4.5, which
is approximately the half of value observed in the RCP8.5 (approximately 17%). Under the RCP8.5 emission scenario
there is a 7-to-23% increase in NPP throughout the basin, with the relative local maxima observed mainly in the coastal
areas of the North Western Mediterranean, Levantine basin, Northern Adriatic Sea, Gulf of Lion, Aegean Sea (similar
results, although with lower rates, were found at the end of the 21st century by Solidoro et al., 2022). Conversely, under
the RCP4.5 scenario, the Adriatic Sea is still characterized by a negative and not significant anomaly (-1%), while for the
rest of the basin the sign of the anomaly is positive and statistically significant, with the greatest values observed in the
North Western Mediterranean (approximately 12%, which is almost half of the variation observed in the RCP8.5
scenario). In both scenarios, a negative anomaly is observed in the Rhodes gyre area, which is extremely weak in RCP8.5.

621      Both negative anomalies are temporally consistent with some convective events taking place in both areas after 2080 and

622      shown in Fig. S4.

623

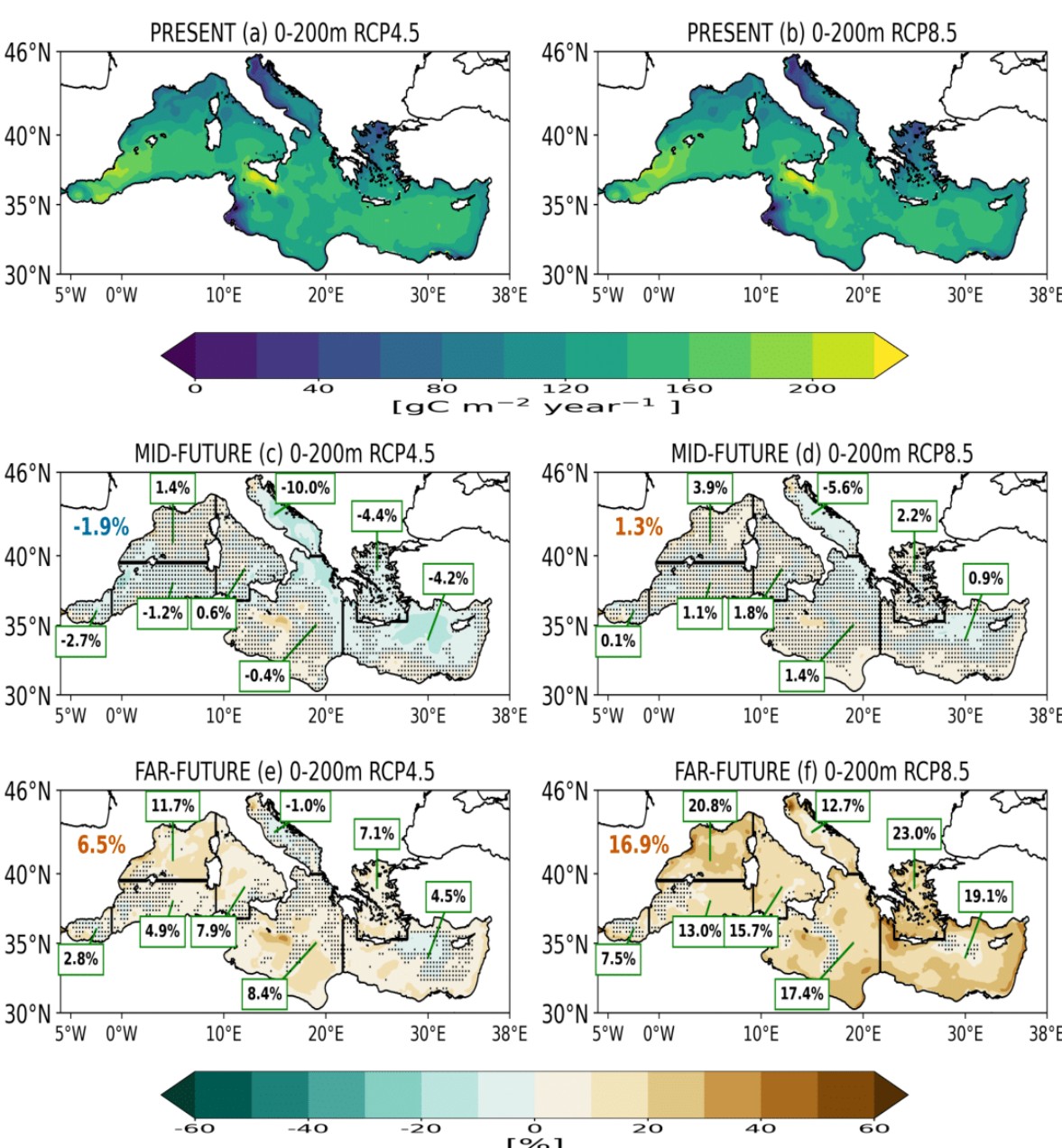

Fig. 11 - Integrated net primary production variation (in gC m$^{-2}$ year$^{-1}$) in first 0-200m in the PRESENT (2005-2020, a,b), and relative climate change signal (with respect to the PRESENT, in units of pH) in the MID-FUTURE (2040-2059, c,d) and FAR-FUTURE (2080-2099, e,f) in the RCP4.5 (left column) and RCP8.5 (right column) scenarios. The Mediterranean average relative climate change signal in each period (with respect to the PRESENT) is displayed by the top-left colored value (blue or dark orange when negative or positive). Values in the green boxes is the average relative climate change signal in each period and in each sub-basin shown in Figure 1. Domain grid

**points where the relative climate change signals are not statistically significant according to a Mann-Whitney test with p<0.05 are marked by a dot.**

As shown by Lazzari et al. (2014) and Solidoro et al. (2022), the overall warming of the water column also results in an increase in community respiration. In agreement with that, Fig. S9 shows the spatial distribution of phytoplankton respiration (RESP) changes in the MID-FUTURE. It is possible to observe some differences with respect to NPP. In both scenarios, there is an overall decrease in the RESP with respect to the beginning of the 21st century, which is approximately equal to -4% in the RCP4.5 and -2% in the RCP8.5. In both scenarios the projected changes are again positive and not statistically significant in the Northern Adriatic, most of the North Western Mediterranean, Central and Southern Ionian and coastal areas of the Levantine basin. As previously observed for NPP, the Adriatic Sea has an overall negative and statistically significant anomaly, as well as the Northern Ionian Sea and the area of the Rhodes gyre. The North Western Mediterranean is the only area where the variation has an opposite sign in two scenarios: it is negative (-1.4%) in RCP4.5 and positive (approximately 1%) in RCP8.5.

In the FAR-FUTURE, the pattern of variation is coherent with that already observed in the NPP (Fig. 11). RESP increases at the end of the 21st century over the entire basin of approximately 5% (11%) in RCP4.5 (RCP8.5). Under the RCP4.5 scenario, the Adriatic Sea, with the northern part as the only exception, and the Rhodes gyre area are still characterized by a negative anomaly while under RCP8.5, the highest values are observed in the North Western Mediterranean (this is also true for the RCP4.5 scenario), Aegean Sea and Levantine basin.

The overall increase in the respiration community has as a consequence the decrease in the organic stock matter in the water column. The temporal evolution of the carbon organic matter standing stock for the 2005-2099 in RCP4.5 and RCP8.5 simulations between 0-100 m and 200-600 m in the whole Mediterranean and in its Western and Eastern basins is shown in Figure 12. The evolution of the carbon organic matter standing stock is similar to that observed in the dissolved nutrients, with a substantially stable signal in the first 30 years of the 21st century and a decrease after 2030. Afterwards, while RCP4.5 shows a recovery at the end of the 21st century, the projected decline in the RCP8.5 is approximately equal to 5 mgC m$^{-3}$. The same dynamics is observed in the intermediate layer, where the decline after the period 2030-2035 is approximately equal to 0.3 mgC m$^{-3}$ for the carbon stock.

Similar dynamics are also observed for plankton (both phyto and zoo, Fig. 13), bacterial biomass and particulate organic matter in the euphotic layer (Fig. 14). In the RCP4.5 simulation for all these biogeochemical tracers, a recovery in the biomass at the end of the 21st century is found and the projected changes are approximately 50% with respect to the RCP8.5 scenario where no recovery is observed. In particular, the decrease of the phytoplankton (zooplankton) biomass is approximately 2 (1.5) mgC m$^{-3}$ and appears to be stronger in the Eastern than in Western basin. Under RCP8.5 the bacterial biomass is projected to decrease at the basin scale by the end of the century by approximately 0.5 mgC m$^{-3}$, by 0.2 mgC m$^{-3}$ in the Western basin and by 0.6 mgC m$^{-3}$ in the Eastern basin. Finally, the decline in particulate organic matter is approximately 1.5 mgC m$^{-3}$ at the basin scale, approximately 1 (2) mgC m$^{-3}$ in the Western (Eastern) basin. In the intermediate layer, the decline of the bacterial biomass in the entire basin is fairly uniform and continuous until the end of the 21st century, with a variation of approximately 0.3 mgC m$^{-3}$ with respect to the beginning of the century. For

the same layer, particulate organic matter declines after the period 2030-2035 but successively the signal remains
substantially stable and, in particular in the Western basin, tends to recover at the end of the century.

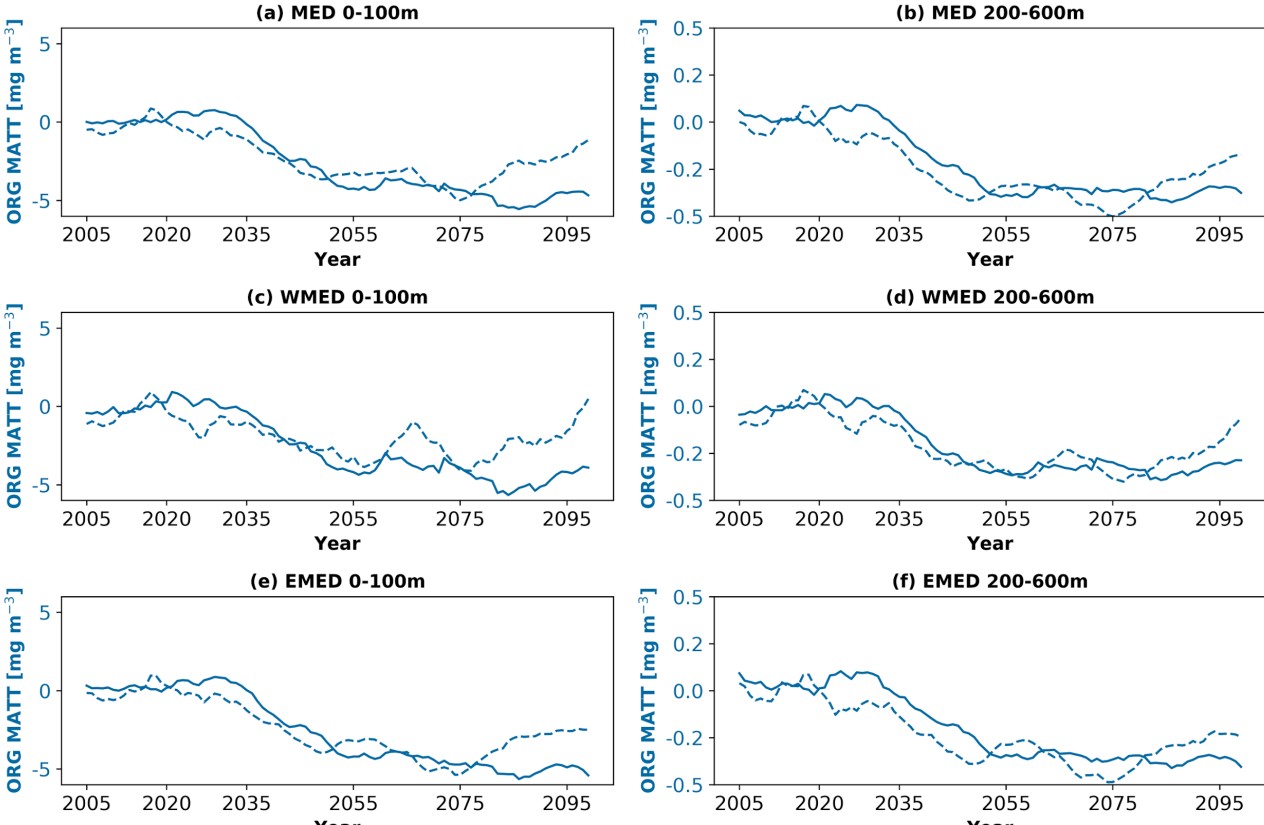


**Fig. 12 - Yearly time-series for the period 2005-2099 of Living/not Living organic Matter (in mg m$^{-3}$) anomalies for the emission**
**scenario RCP8.5 (solid line) and RCP4.5 (dashed line) in the Mediterranean Sea (MED, a-b), Western Mediterranean (WMED,**
**c-d) and Eastern Mediterranean (EMED, e-f) for the layer 0-100 m (left column) and 200-600 m (right column) for the 2005-**
**2099 period. The yearly time series have been smoothed using 10-years running mean.**


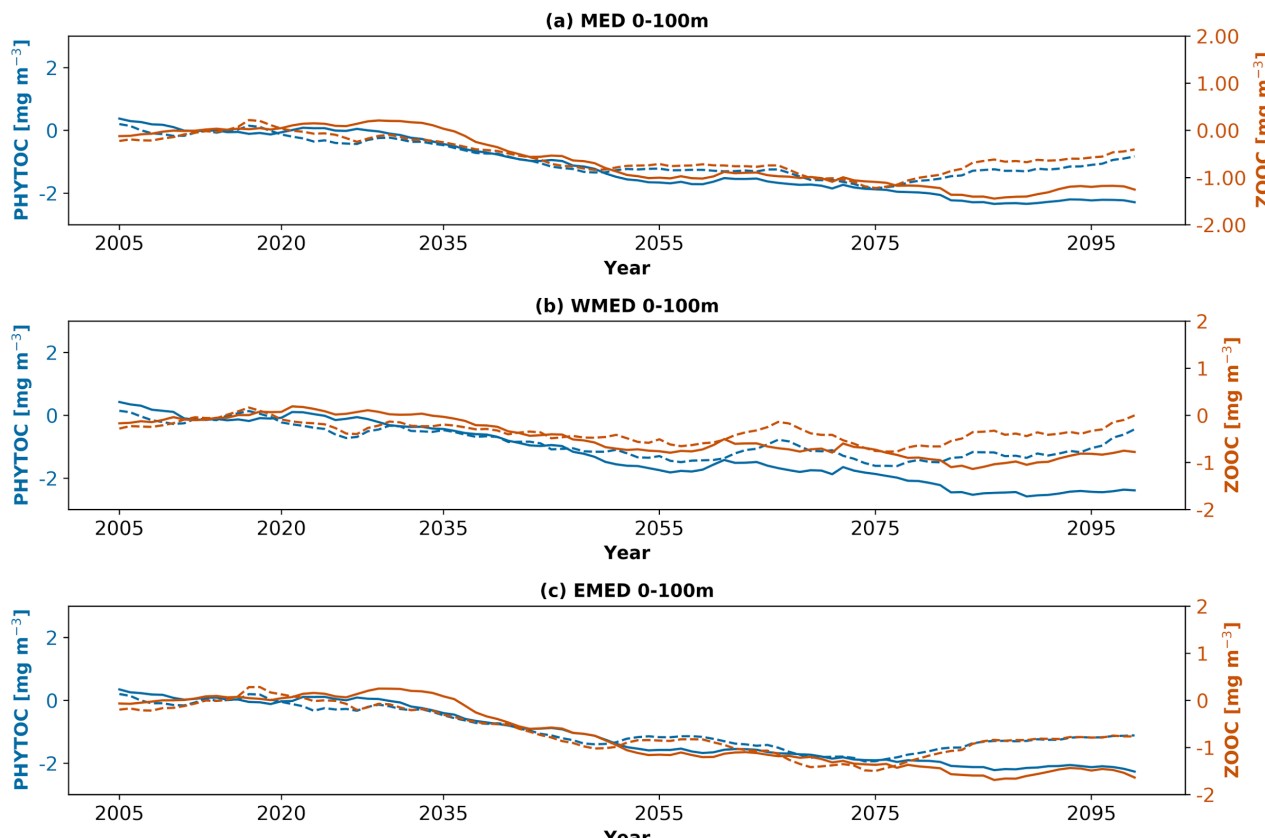

**Fig. 13 - Yearly time series of Phytoplankton biomass (blue, in mg m⁻³) and Zooplankton (dark orange, in mg m⁻³) anomalies for the emission scenario RCP8.5 (solid line) and RCP4.5 (dashed line) in the Mediterranean Sea (MED, a), Western Mediterranean (WMED, b) and Eastern Mediterranean (c) for the layer 0-100m and for the 2005-2099 period. The yearly time series have been smoothed using 10-years running mean.**

In the two scenarios, in both MID-FUTURE and FAR-FUTURE, the areas most affected by the statistically significant decline of phytoplankton (Fig. S10) and zooplankton (Fig. S11) biomasses are mainly the sub-basins of the Eastern Mediterranean Sea, namely the Ionian Sea (mainly its Northern part), the Adriatic Sea (except for its Northern part), the Aegean Sea and the Levantine basin (in particular the Rhodes gyre area) and the Tyrrhenian Sea (only for the phytoplankton). Moreover, the negative anomaly in the area of Rhodes gyre is spatially coherent with the anomalies observed in the case of NPP and RESP, consequences of the vertical convection phenomena in the area. Conversely, positive but not statistically signals for both variables can be observed only at the local scale in the Strait of Sicily and along the coast of the North Western Mediterranean (spatially coherent with the positive variations of the PO4 discussed in section 3.3 and in both cases not significant).

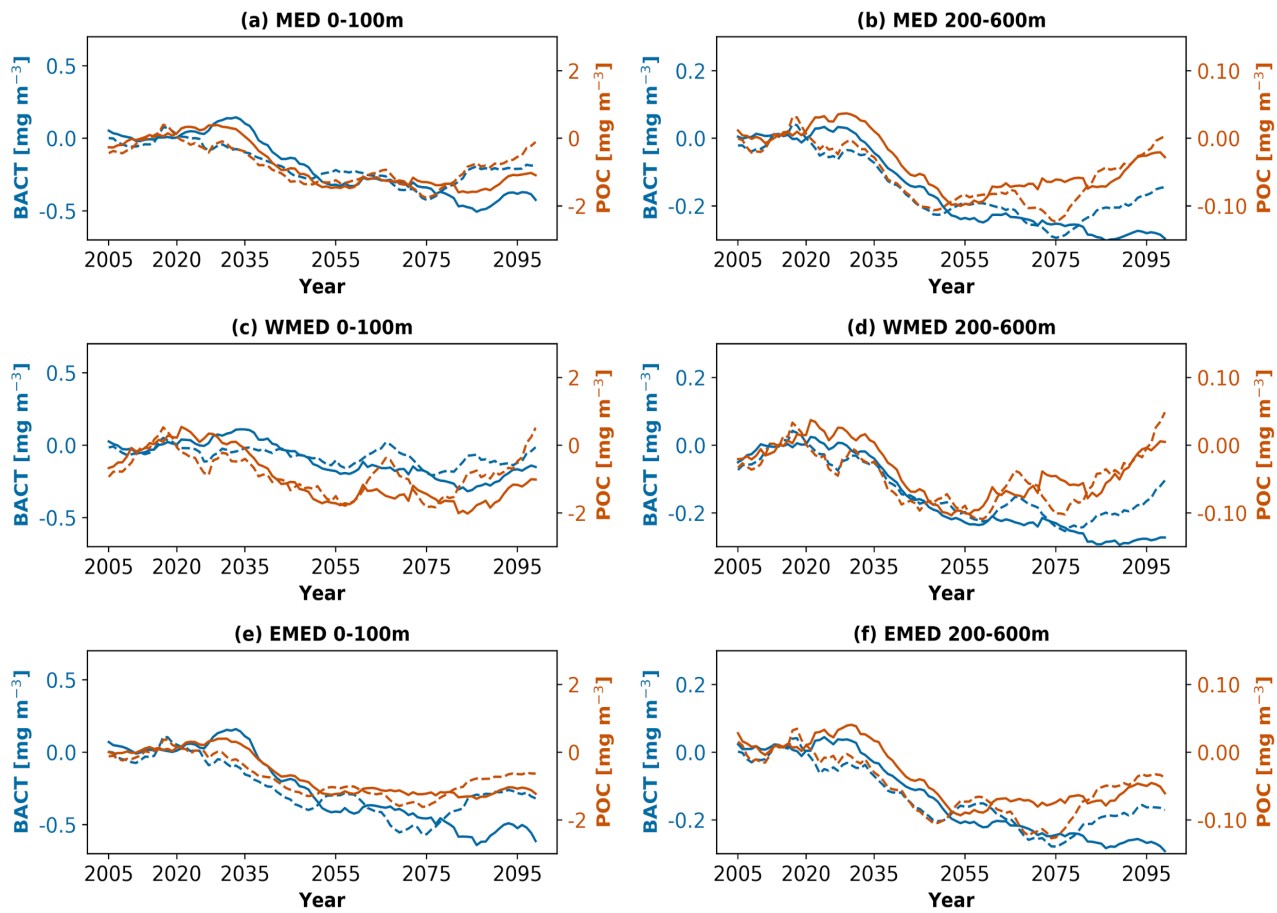


**Fig.14- As Fig.8 but for bacterial biomass (blue, in mg m⁻³) and particulate organic matter (dark orange, in mg m⁻³)**

Also in the case of bacterial biomass (Fig. S12) and particulate organic matter (Fig. S13) the decline, along the 21st
century, will mostly affect the euphotic and the intermediate layers of the Eastern basin, in both MID- and FAR-FUTURE,
with relative maxima observed in the Levantine basin (around 33.5% in RCP4.5 and 50% in the RCP8.5 scenario). This
decline is related to an increase of the respiration at community level, as observed for phytoplankton (Fig. S9). However,
there are some exceptions to the general decline of the bacterial biomass and particulate organic matter in the basin. For
example, in the Adriatic Sea, under scenario RCP8.5, the decrease of the bacterial biomass with respect to the beginning
of the century is only 1% with a slight positive anomaly appearing in the Southern Adriatic at the end of 21st century (not
statistically significant, Fig. S12). In the case of particulate organic matter, the Strait of Sicily and the Northern Adriatic
Sea are characterized by a permanent positive signal in both layers and scenarios as observed before for $PO_4$, also in this
case not statistically significant. Moreover, in RCP4.5 simulation, in the FAR-FUTURE period, the North Western
Mediterranean shows an increase of the particulate organic matter content in the euphotic and intermediate layers.

**3.4 Spatial and temporal evolution of dissolved inorganic carbon (DIC) and pH**

A basin-wide continuous increase in DIC is projected until the end of the 21st century, with a stronger signal observed in
the RCP8.5 scenario (Fig. 15), and more specifically, in the Eastern part of the Mediterranean basin. In fact, in the euphotic
layer, the increase in DIC with respect to the beginning of the century is approximately 150 $\mu$mol kg⁻¹ under RCP8.5 in

the Eastern basin, while it is approximately 120 $\mu$mol kg$^{-1}$ in the Western basin. Additionally, in the intermediate layer, DIC increases by approximately 120 $\mu$mol kg$^{-1}$ with respect to the beginning of the century: this value is approximately the same for both the Western and Eastern basins and is double with respect to that observed in the RCP4.5 scenario.

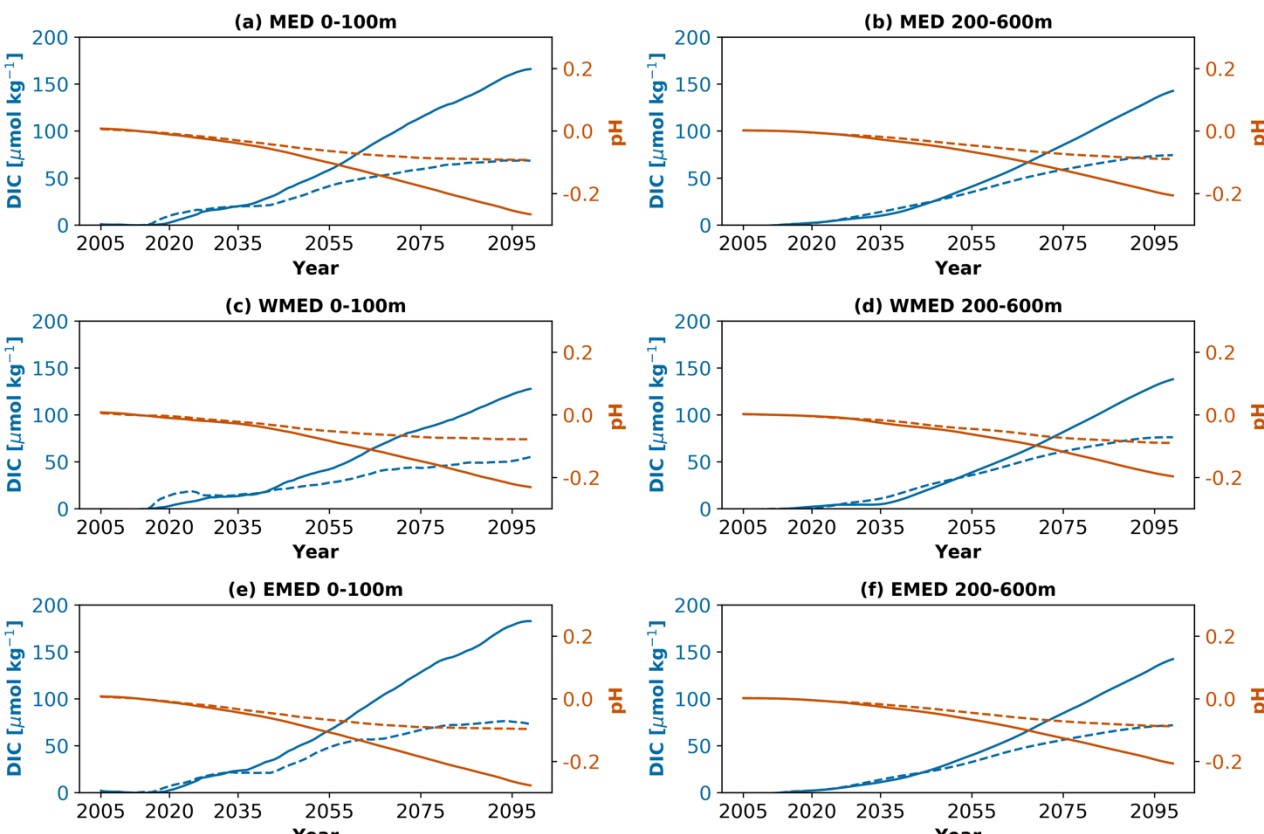

**Fig. 15 - as Fig.14 but for Dissolved Inorganic Carbon (blue, in μmol kg$^{-1}$) and pH (dark orange)**

Although community respiration can play a role in the increase in DIC, in the Mediterranean region a predominant mechanism is represented by the air-sea $CO_2$ exchange (D'Ortenzio et al., 2008; De Carlo et al., 2013; Hassoun et al., 2019; Wimart-Rousseau et al., 2020). In fact, looking at the terms controlling the DIC increase, the air-sea $CO_2$ exchange shows an almost balanced condition in the present-day (D'Ortenzio et al., 2008; Melaku Canu et al., 2015), and an increase throughout the 21st century as a consequence of the increase in atmospheric $CO_2$ (Fig. 16, a,b). The $CO_2$ flux increase is almost linear and is equal in the two scenarios until 2050. Then, the RCP4.5 scenario shows a smoothing in the second half of the century, which is consistent with the reduced atmospheric emissions, while the linear increase persists under RCP8.5 (Fig. 16, a,b).

The two main Mediterranean sub-basins behave quite differently: the $CO_2$ air-sea sink is three times greater in the Western part compared to the Eastern part, reflecting the influence of both DIC and temperature spatial gradients (i.e., higher values of DIC and temperature in the Eastern basin). In order to assess the temperature and DIC contributions to the pCO$_2$ temporal evolution, the carbonate system equations of the BFM model have been solved in offline mode, keeping constant, alternatively, temperature and DIC concentration. The increase in the temperature has been shown to account

for 25% of the total increase in the $pCO_2$. The remaining part of the $pCO_2$ increase can be ascribed to the DIC
concentration increase. In the Western part, a less pronounced temperature effect (i.e., temperature increases slower in
the Western part) causes an undersaturation condition of $pCO_2$ (i.e., $pCO_2^{sea}$ lower than $pCO_2^{atm}$ values) compared to the
Eastern conditions, triggering a much higher $CO_2$ absorption in the Western Mediterranean.

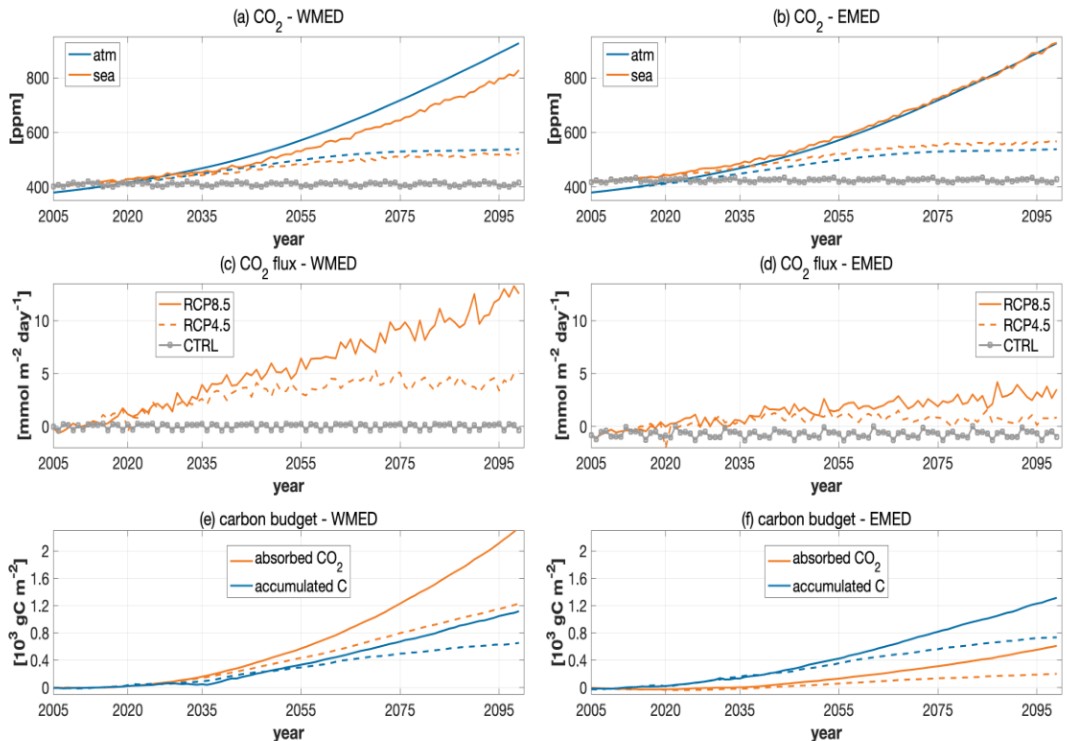



**Fig.16 - Time series of atmospheric and marine $pCO_2$ (a,b), $CO_2$ air-sea exchange (c,d) and cumulative $CO_2$ absorbed and**
**accumulated in the water column (e,f) in the Western (a,c,e) and Eastern (b,d,f) Mediterranean Sea. Two scenarios RCP4.5**
**(dashed line) and RCP8.5 (continuous line) and control simulation (CTRL, gray line) are reported.**

As a result of the air-sea $CO_2$ sink, for example the RCP8.5 scenario shows a steady DIC accumulation after 2030 with
values of more than 2 $\mu$mol kg$^{-1}$ year$^{-1}$ in the first 600 m (500 m) of the water column for the Western basin (Eastern
basin; Fig. 17).

The increase in DIC in the upper layer is approximately 1.5% and 2.5% in the Western and Eastern basins, respectively,
in the MID-FUTURE, and 5% and 7% in the FAR-FUTURE (Fig. S14). In the 200-600 m layer, the DIC increase follows
the same pattern as that in the upper layer, but with smaller changes (i.e., approximately 1-2% less). Then, while the DIC
increase does not impact the water column below 1200 m in the Western basin, DIC still accumulates until 2000 m in the
Eastern basin at a rate of almost 0.5 $\mu$mol kg$^{-1}$ year$^{-1}$ (Fig. 17). Occasional events of deep transport of DIC can be
recognized (e.g. around the years 2035, 2045, 2085 and 2095, similar to what observed in the case of oxygen in Fig.9)
and the water column results enriched down to 1000-1500 m with a rate of approximately 1 $\mu$mol kg$^{-1}$ year$^{-1}$. In the surface
layer (i.e., first 50-100 m), the interannual variability of atmospheric conditions (i.e. specific annual wind and temperature
seasonal cycles triggering the $CO_2$ fluxes) and the winter mixing produce an irregular succession of positive and negative
annual changes, which can partially hide the long-term effect of the increase in atmospheric $pCO_2$. Thus, the cumulative
sum of the $CO_2$ absorbed through air-sea exchanges and of the carbon accumulated in the water column (Fig. 16, e,f)
highlight the different behavior of the two main sub-basins. The Western basin absorbs much more atmospheric $CO_2$ than
the Eastern basin, with even larger differences in the RCP8.5 scenario. By the end of the RCP8.5 scenario, 1.8 PgC of
atmospheric $CO_2$ sink in the Western basin while only 1 PgC in the Eastern basin are observed, consistent with the
estimates of Solidoro et al. (2022).

However, the fate of the absorbed carbon is quite different: the Western basin during the 21st century (RCP8.5 scenario)
accumulates only 0.85 PgC, while 1.7 PgC are retained in the water column of the Eastern basin. As shown in Figure 16
(lower panel) for the RCP8.5 scenario, the Eastern basin accumulates almost 2 moles of carbon for each atmospheric $CO_2$
mole absorbed (up to 3 in the RCP4.5), while it is less than 0.5 for the Western basin. The different efficiency is eventually
triggered by the thermohaline circulation change: the Western Mediterranean carbon is partly exported to the Northern
Atlantic Ocean, while an increased quota of carbon input from rivers and across the Sicily channel are retained in the
Eastern basin together with the atmospheric $CO_2$ sink after the weakening of the thermohaline circulation (Fig.4). The
RCP4.5 scenario shows similar dynamics to RCP8.5, with rates of $CO_2$ absorption (Fig. 16) and of DIC accumulation
almost halved, and the impact of the interannual variability on surface layer dynamics much more amplified (not shown).
As a result, the total sequestered atmospheric $CO_2$ equals to 0.8 and 0.25 PgC in the Western and Eastern basins, while
the increases of the carbon pool are 0.5 PgC and 0.9 PgC, respectively.

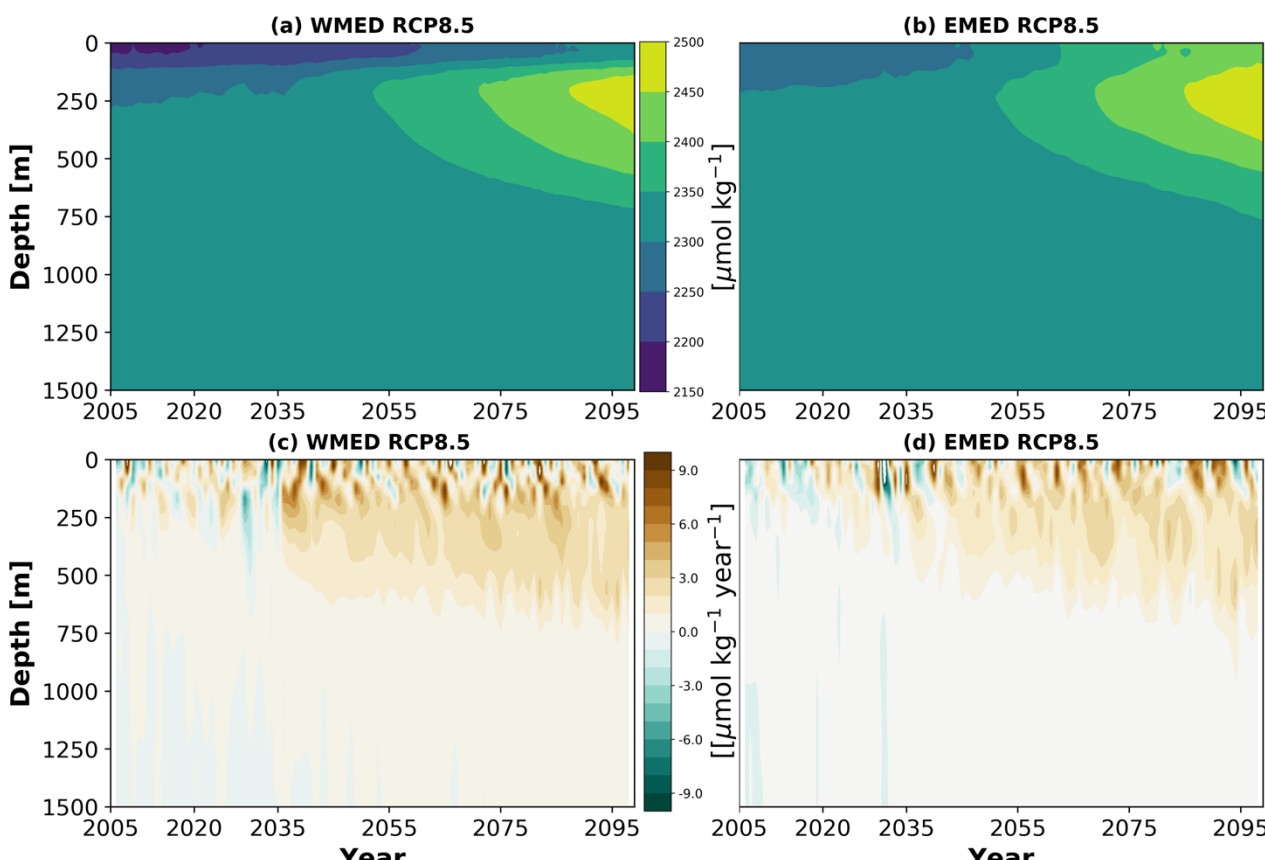


**Fig. 17 - Hovmoeller diagram of DIC (µmol kg$^{-1}$, panel a,b) and annual rate of change of DIC (µmol kg$^{-1}$year$^{-1}$, panel c,d) in the Western (a,c) and Eastern (b,d) Mediterranean Sea in RCP8.5 scenario.**

Consequently, to the $CO_2$ invasion and DIC increase, the change in the carbonic acid equilibrium causes a generalized decrease in pH, as also shown in Solidoro et al. (2022) in the case of the A2 scenario. The change in pH, which is statistically significant everywhere and very well anti-correlated in time and space with the DIC change (on the basin scale the correlation coefficient is lower than -0.90 with p<0.05; Fig.S14) and almost similar in both Western and Eastern Mediterranean (as already projected by Goyet et al, 2016), is approximately by the end of the century equal to 0.1 in the RCP4.5 and 0.25 pH units in the RCP8.5 scenario (Fig. 18). The largest decreases in pH are projected in both scenarios in the upper layer of the North-Western Mediterranean, Tyrrhenian Sea, Adriatic Sea and Aegean Sea and in the 200-600 m layers of the Tyrrhenian Sea, Ionian Sea and Aegean Sea in the FAR-FUTURE (Fig. 18).

pH

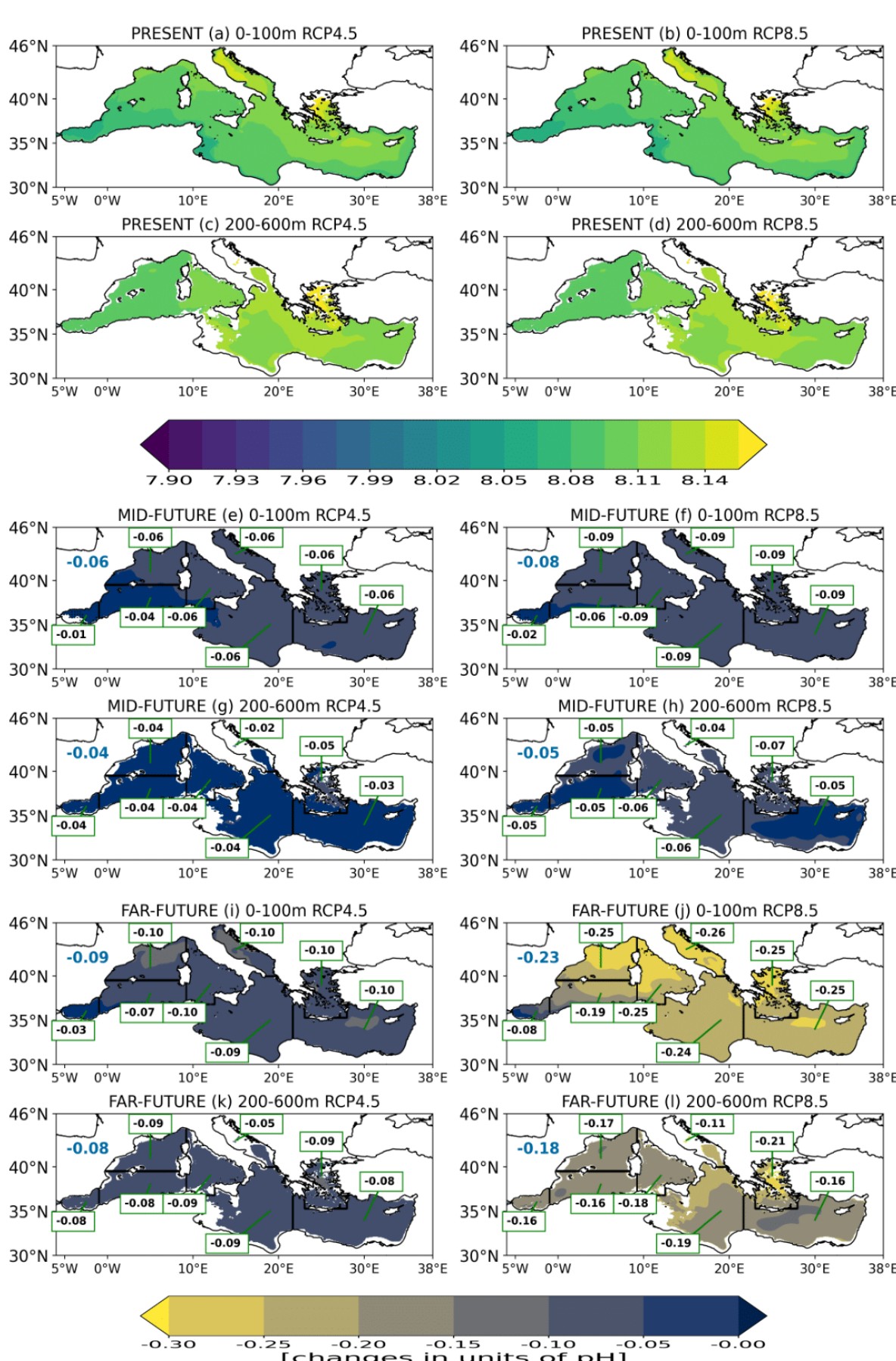

**Fig. 18 –pH in the layers 0-100m and 200-600m in the PRESENT (2005-2020, a,b,c and d), and relative climate change signal**
**(with respect to the PRESENT, in units of pH) in the MID-FUTURE (2040-2059, e,f,g and h) and FAR-FUTURE (2080-2099,**
**i,j,k and l) in the RCP4.5 (left column) and RCP8.5 (right column) scenarios. The Mediterranean average relative climate**
**change signal in each period (with respect to the PRESENT) is displayed by the top-left colored value (blue or dark orange**
**when negative or positive). Values in the green boxes is the average relative climate change signal in each period and in each**
**sub-basin shown in Figure 1. Domain grid points where the relative climate change signals are not statistically significant**
**according to a Mann-Whitney test with p<0.05 are marked by a dot.**

**4. Discussions and Conclusions**

In this study, the coupled physical-biogeochemical model MFS16-OGSTM-BFM is used to simulate the biogeochemical
dynamics of the Mediterranean Sea during the 21st century under the two emission scenarios RCP4.5 and RCP8.5, and
to assess some climate-related impacts on the marine ecosystems of the basin.

To the best of the authors' knowledge, this work is the first one that analyzes long-term eddy-resolving projections of the
biogeochemical dynamics of the Mediterranean Sea under two different emission scenarios. In fact, the horizontal and
vertical resolution ($1/16°$ and 70 vertical levels) of the long-term projections here analyzed is higher than that of previous
works available in the scientific literature that focuses on the area (e.g Lazzari et al., 2014; Macias et al., 2015; Richon et
al., 2019; Pagès et al., 2020; Solidoro et al., 2022). Moreover, the majority of the recent scientific works discussing the
impacts of climate change on the biogeochemical dynamics of the Mediterranean Sea are based on the analysis of
simulation that considered the worst-case emission scenario (A2 or RC8.5; Moullec et al., 2019; Richon et al., 2019;
Pagés et al., 2020; Solidoro et al., 2022).

The use of eddy-resolving resolution and of a higher vertical resolution allows a more detailed representation of the
vertical mixing and ocean convection processes, which play a fundamental role in the ventilation of the water column and
in the nutrient supply into the euphotic layer of the basin (Kwiatkowski et al., 2020). Moreover, the use of a $1/16°$
horizontal resolution for the projections has allowed to resolve, identify and characterize, for the first time, spatial
gradients existing in the same sub-basins (such as in the Adriatic Sea) or between coastal and open ocean areas (such as
in the North Western Mediterranean). A more detailed representation of the spatial distribution of the projected changes
and of their statistical significance for different biogeochemical tracers and properties represents a clear advantage for the
future assessment of climate change impacts on specific organisms, habitats or target areas, also at sub-basin scale.

The analysis of the thermohaline properties and circulation of the Mediterranean Sea under emission scenarios RCP4.5
and RCP8.5 found different levels of warming of the water column and weakening of the thermohaline circulation cell,
with different parts of the basin being characterized by contrasting saltening and freshening conditions as a function of
the considered scenario. Moreover, different levels of weakening of the open ocean convection in the most important
convective areas of the basin are projected, with the only exception of the Aegean Sea, where episodes of deep convection
similar to the EMT could be observed at the end of the 21st century (see also Adloff et al., 2015). All the projected
changes are in agreement with those already depicted in recent model studies (e.g. Somot et al., 2006; Adloff et al., 2015;
Waldman et al., 2018; Soto-Navarro et al., 2020).

A comparison of the model outputs with available data in the present climate, together with previous studies performed
with the same model system, support the conclusion that the coupled model MFS16-OGSTM-BFM has a reasonably good
ability in reproducing the main biogeochemical features of the Mediterranean Sea and can be used as a tool for assessing
the future biogeochemical dynamics of the basin and its changes in response to climate change. The use of the bias-
removing protocol is often advocated as a good practice in climate studies, but rarely implemented in biogeochemical or
ecosystem projections (e.g., Solidoro et al., 2022) and it adds further robustness to our results.

Our projections for the biogeochemical tracers and properties at the end of the 21st century shows several signals (see
Table SP1 for a synthetic overview) that are mostly in agreement with previous studies, at least with those based on the
use of the worst-case emission scenarios. The magnitude of the projected changes has been shown to be, in general,
scenario-dependent with the largest deviations from the present climate state observed in the RCP8.5 emission scenario
(Table SP1). On the other hand, the analysis of the projections under RCP4.5 found in most of the biogeochemical
variables (for example dissolved nutrients and biomasses) by the end of the 21st century a tendency to recover the values
observed in the present climate (Table SP1).

As shown in the previous sections, our simulations, by covering also the RCP4.5 scenario, highlight how an intermediate
greenhouse emission scenario produces results that are not simply an average between the present condition and the
RCP8.5, but (at least for some variables) something quantitatively different. For example, the temporal evolution of pH
(Fig.15) is similar in two scenarios in the first 30 years of the 21st century. Conversely, after 2050, pH undergoes a
substantial decrease under RCP8.5 while it remains almost stable under RCP4.5 with a final projected variation lower
than the half with respect to the worst-case scenario. This supports the idea - possibly based on the existence of a certain
buffer capacity and renewal rate in a system like the Mediterranean Sea - that the implementation of policies of reducing
$CO_2$ emission could be, indeed, effective and could contribute to the foundation of ocean sustainability science and
policies.

The decline in the dissolved nutrients at the surface under RCP8.5 scenario is comparable with that observed in Richon
et al. (2019). However, they project an overall increase in the concentration of both nutrients at the surface after 2050,
which is ascribed by the authors to river and Gibraltar inputs that are not constant over time (as in our case) but are based
on a global climate scenario simulation. As highlighted by Richon et al. (2019), the sensitivity of the biogeochemical
fluxes at the river loads and Gibraltar exchanges is of paramount importance, and surely worthy of further investigation.
Nevertheless, the increase in the concentration of nutrients in the intermediate layers of both the Western Mediterranean
and Levantine basin can be also traced back to the reduced vertical mixing resulting from the increase in the vertical
stratification (Somot et al., 2006; Adloff et al., 2015; Richon et al., 2019).

Different levels of increase in the net primary production and respiration are projected in both scenarios although many
recent studies in the Mediterranean region have shown a different response of integrated net primary production to climate
change in both Western and Eastern basins (e.g. Macias et al., 2015; Moullec et al., 2019; Pagès et al., 2020). In fact, this
response may vary according to the sensitivity of the assumptions (model equations) for primary production and recycling
processes to changes in temperature (Moullec et al., 2019). In the BFM model temperature regulates most of the metabolic
rates with a Q10 formulation (Vichi et al., 2015). The increase in net primary production is a consequence of such
dependence. In other studies (Eco3M-Med model; Pages et al., 2020) organisms are always optimally adapted and no
temperature dependence is accounted for in the physiology. This different parameterization could be connected to the
different results in terms of trends; in fact, the scenarios based on the Eco3M-Med model results in a reduction of net
primary production. In this case surface nutrient reduction, rather than temperature, affects the net primary production
trend producing a decrease. The relative impact of different drivers (nutrient supply versus organism's adaptation to
average water temperature) could be explored with dedicated sensitivity experiments.

Our projections of net primary production and biomass dynamics show how different levels of warming of the water
column and consequent stratification have a direct impact on the ecosystem functioning by increasing the metabolic rates.
Similar to the results obtained in Lazzari et al. (2014) and Solidoro et al. (2022), the increase in metabolic rates augments
both primary production and respiration, but with the net effect of reducing living and non-living particulate organic
matter, as suggested from theoretical considerations in O'Connor et al. (2011). The decoupled formulation of carbon
uptake and net growth in the BFM model induces a further mechanism related to how carbon is channeled in the food
web. In fact, the decrease in biomass is partially compensated by an increase in dissolved organic matter production in
the basin by the end of the century (Solidoro et al., 2022; results not shown here).

Basin-wide deoxygenation tendencies are found in both scenarios and are comparable to trends observed on the
Mediterranean scale by Powley et al. (2018) and, under RCP8.5, on the global scale by CMIP6 simulations (Coupled
Model Intercomparison Project Phase 6; Kwiatkowski et al., 2020). The former, using a box model, found a decrease in
the oxygen content of the intermediate layer in the range of 2-9% as a consequence of different projected changes in the
solubility (due to the temperature increase) and in the thermohaline circulation of the basin. Furthermore, the projections
show that, in both our scenarios, deoxygenation is higher in the Eastern than the Western basin, where the Atlantic
boundary condition might have dumped the deoxygenation trend, and in several coastal areas such as the Northern
Adriatic (until -25 mmol $m^{-3}$). As also observed by Powley et al. (2018), the main driver of deoxygenation is the change
in solubility, whereas changes in the circulation (i.e., weakening of the thermohaline circulation) should not substantially
affect deep ventilation, and it is unlikely, even in the worst-case scenario, to reach hypoxia conditions in the deep layer
of the basin by the end of the century. On the other hand, the greatest threat for the oxygen water content might be linked
to the combination of surface warming and faster respiration processes in the coastal areas of the basin which could result
in lower oxygen conditions and, thus, alteration of the local marine ecosystem functioning and structures (Bindoff et al.,

906     2019).


An increase in the dissolved inorganic carbon content and acidity of the water column (Solidoro et al., 2022) is found in
both scenarios. The overall accumulation of $CO_2$ in the basin resulted in an acidification of the Mediterranean water with
a decrease in pH of approximately 0.23 units in the worst-case scenario, which is slightly lower than the 0.3 projected on
a global scale (Kwiatkowski et al., 2020) and lower than the value provided in Goyet et al. (2016), who projected, using
thermodynamic equations of the $CO_2$/carbonate system chemical equilibrium in seawater, a variation of 0.45 pH units in
the basin under the worst SRES case scenario (and 0.25 pH units in the most optimistic SRES scenario). However, this
last estimate probably tends to overestimate the future acidification of the basin, as it does not consider the decrease in
the exchanges and the penetration of $CO_2$ across the ocean-atmosphere interface due to the warming of the water column
(MedECC, 2020).
This difference in the response to climate change between the Western and Eastern basins has been also observed for the
dissolved inorganic carbon accumulation and reflects indeed different factors such as the different ventilation and
residence time of water masses in the two basins as well as the exchanges in the Strait of Gibraltar (e.g. Alvarez et al.,
2014; Stöven and Tanhua, 2014; Cardin et al., 2015; Hassoun et al., 2015). Results show that, in both scenarios, the
Western basin, while adsorbing greater quantities, accumulates only a half of the atmospheric carbon stored by the Eastern
basin because in the former the carbon is partly exported to the Northern Atlantic Ocean, while in the latter, it is also
affected by a more intense reduction of the thermohaline circulation and therefore in the vertical transport processes, the
carbon is retained together with the atmospheric $CO_2$ sink. Additionally, in our case, the use of a high resolution for the
biogeochemical projections has allowed to show that in many coastal areas the observed acidification is lower by
approximately 8% with respect to the open ocean due to damping effects of ALK input from the rivers (not shown here).
The decline in many biogeochemical tracers and properties in the euphotic layer begins in the 2030-2035 period, in
correspondence to the weakening of the thermohaline circulation in the basin (Fig. 4), and it is particularly marked in the
Eastern basin. This shows that the modification of the circulation resulting from future climate scenarios has substantial
effects on the biogeochemical properties of the basin. Changes in the thermohaline circulation of the basin also explain
the increase in the nutrient concentration in the intermediate layer of the Levantine basin, which is a result of the
weakening of the westward transport of nutrients through the Strait of Sicily (Fig.S5).
Similar to all previous modelling cited studies (e.g Lazzari et al., 2014; Macias et al., 2018; Richon et al., 2019; Pagès et
al., 2020), some sources of uncertainties for our projections need to be considered. As discussed before, MFS16
adequately reproduces the distribution of key physical properties and the thermohaline circulation of the basin. On the
other hand, recent studies based on multi-model ensembles (Adloff et al., 2015; Richon et al., 2019; Soto-Navarro et al.,
2020) have suggested that atmospheric forcing and boundary conditions can strongly affect the dynamics of the basin,
particularly the vertical mixing, which plays a primary role in the distribution of nutrients in the euphotic layer, therefore
affecting the dynamics of low trophic levels. Additional sources of uncertainties in the modelling framework can be traced
back to the BFM biogeochemical model. For instance, in the present climate the model tends to overestimate the chl-*a* at
the surface and, even more, the oxygen concentration below 200 m (section 3.1). These overestimations can be propagated
by the integration into the future projections. However, the conclusions of the present work should not be significantly
affected by that because, at the same time, the CTRL simulation is also removed from both the scenario simulations.
Moreover, the signs of the projected changes (not their absolute values) result from different physical and biogeochemical
processes (e.g., temperature and respiration increase, weakening of the thermohaline circulation, increase in the
stratification) which are linked to the climate forcing and are independent from model uncertainties that generate the
biases discussed in section 3.1.
Furthermore, the set-up of the boundary conditions, namely the atmospheric deposition at the surface, the rivers nutrient
loads and the vertical profiles in the Atlantic boundary can be very critical, especially in the land-locked Mediterranean
basin. Atmospheric deposition is an important source of nutrients for the basin and it has been shown that the
biogeochemical dynamics of the Mediterranean Sea is influenced by aerosol deposition (e.g. Richon et al., 2018, 2019),
especially during periods of stratification. The projected lower nutrient supply from sub-surface waters caused by climate-
driven stronger stratification, could likely increase the importance of the atmospheric deposition as a source of nutrients
for the euphotic layer (Gazeau et al., 2021). Thus, possible future changes in the deposition of aerosols could influence
the biogeochemistry of the basin and the nutrients concentration at the surface as projected for the 21st century and
depicted in Section 3.3. However, in both RCP4.5 and RCP8.5 simulations, a present-day phosphate and nitrogen
deposition is used. Potential improvements will be achieved indeed by the inclusion of more accurate deposition
information derived from CMIP6 global estimates for the 21st century (O'Neill et al., 2016).

Similarly, the lack of river nutrient load projections under the prescribed emission scenarios can affect the projected
nutrient budget of the Mediterranean basin. A climatology derived from the Perseus project (see Section 2.3) is here
adopted, which is, to our knowledge, the most reliable information. Indeed, it is reasonable to assume that land-use and
runoff changes might impact future nutrient loads, although the magnitude and even the sign are presently unknown. Our
river runoff was based on projections (Gualdi et al., 2013; Section 2.1) which estimated an average decrease by the end
of the 21st century. Thus, the increase of nutrients observed in Fig. 9 and Fig. 10 in the Northern Adriatic and several
coastal areas of the Western basin can be partially related to the mismatch between a constant nutrient load and a
decreasing runoff. However, it might be worth remembering that the amount of nutrients entering the basin through its
boundaries ultimately depends on the economic policies and land use/coverage scenarios and therefore they may be
intrinsically subjective.

With DIC as the only exception, for the 21st century, a single vertical profile based on the present-day condition data is
used and no future evolutions are considered for the boundary conditions at the Strait of Gibraltar. If this approach allows
to point out the effects of the changes in the basin circulation on the nutrient budgets, it could miss the influence on
nutrients or other biogeochemical properties of a possible different future evolution of the exchanges in the Strait of
Gibraltar due to changes in the tracer concentrations in the Atlantic Ocean. Moreover, the use of the same Atlantic
boundary conditions for the two scenarios (section 2.3) could have led to an underestimation of a potential difference
between the two scenarios in the areas most influenced by the Atlantic boundary (e.g. Alboran Sea and South Western
Mediterranean). Recent physical simulations have shown an increase of 3.7% in the surface flow at the Strait of Gibraltar,
which could imply an increase in the inflow of nutrients in the surface layer at Strait of Gibraltar (Richon et al., 2019;
Pagès et al., 2020), thereby eventually damping the decrease in the nutrient concentration at the surface projected for the
21st century. As previously observed, this could explain the observed differences among different studies that analysed
future projections of the biogeochemistry of the basin.

To conclude, the methodology and results here presented, provide a robust picture of the evolution of the Mediterranean
Sea biogeochemistry for the 21st century. Clearly, the new generation of Regional Earth System Coupled Models
(RESM), with eddy-resolving ocean models such as the one exploited here, may partially reduce the limitations of using
external (and possibly misaligned) sources of information for atmospheric and land input to the ocean. Indeed, by directly
resolving the coupling between the Mediterranean Sea, the regional atmospheric domain and the hydrological component,
a regional earth system coupled model (e.g., as in Sitz et al., 2017, and Reale et al., 2020a) allows the simulation of the
different components of the climate system at the local scale, including aerosol and river loads.

**Acknowledgements**

M. Reale has been supported in this work by OGS and CINECA under HPC-TRES award number 2015-07 and by the project FAIRSEA (Fisheries in the Adriatic Region - a Shared Ecosystem. Approach) funded by the 2014 - 2020 Interreg V-A Italy - Croatia CBC Programme (Standard project ID 10046951). This work has been partially supported by the Italian PRIN project ICCC (Impact of Climate Change on the biogeochemistry of Contaminants in the Mediterranean Sea). The CINECA consortium is acknowledged for providing the computational resources through the IscraB project "Scenarios for the Mediterranean Sea biogeochemistry in the 21st century" (MED21BIO) and the IscraC DYBIO, EMED18 and MEDCLI16. This study has been conducted also using E.U. Copernicus Marine Service Information. The authors acknowledge Dr. Abed El Rahman Hassoun, an anonymous Reviewer and the Associate Editor Jean-Pierre Gattuso for providing suggestions and criticisms that greatly improved the study.

**Data availability**

Data produced in the numerical experiments are available through the portal dds.cmcc.it for both physical (https://dds.cmcc.it/#/dataset/medsea-cmip5-projections-physics) and biogeochemical (https://dds.cmcc.it/#/dataset/medsea-cmip5-projections-biogeochemistry) components.

**Author contribution**

GC, PL, SS, MR and CS conceived the study. They designed the experiments together with TL. MR, GB and TL performed the numerical simulations. MR, GC, SS, TL and PL performed the analysis of the simulation results. MR prepared the first draft of the manuscript under the supervision of SS, GC, PL and CS and with the contribution from all the authors. All the authors discussed the results and contributed to the revision of the manuscript.

**Competing interest**

The authors declare that they have no competing interests.

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
