# Peer review of "Acidification, deoxygenation, nutrient and biomasses decline in a warming Mediterranean Sea"

_Biogeosciences, 2021_

## Author Comment (AC2)

**Author responses to Reviewer#2 comments for the manuscript: "Acidification, deoxygenation, nutrient and biomasses decline in a warming Mediterranean Sea"**

**February, 18th 2022**

This paper describes new high-resolution coupled (physic and biogeochemistry) simulations made under the RCP 4.5 and 8.5 scenarios for the Mediterranean Sea (MED). This work is in line with the studies that have been carried out over the last ten years to assess the effects of climate change on the circulation and biogeochemistry of the MED. The main improvement here is the use of a higher spatial resolution than the previous studies. I acknowledge the huge amount of work and computational time that is required to run such simulations and I do think that this study is a step forward to a better understanding of the effect of climate change on the MED. However, despite the overall good quality of this work, this paper could be improved.

We thank the Reviewer #2 for their positive feedback and for providing detailed comments and suggestions, which will be considered to improve the manuscript. Reviewer's comments are in bold, authors' responses are in normal font, italicized where they quote proposed changes to the manuscript.

The use of higher resolution simulations could significantly improve our understanding of the MED in the context of climate change, but the authors don't really highlight the interest of such high resolution in the manuscript. The authors propose an analysis based on two averaged integrated depths 0-100 m and 200-600 m for 8 Mediterranean sub-basins losing in my opinion the opportunity to fully take advantage of their model resolution. Furthermore, the discussion as proposed here lacks rigorous comparison and discussion with the existing literature which could emphasize more the importance of high resolution. I also have concerns about the methodology, for example (but not only) the choice of the reference periods (2005-2020) that have been taken into account including the beginning of the scenarios, and the choice of allowing only the dissolved inorganic carbon to change at Gibraltar Strait.

We thank the Reviewer for raising different interesting points. We are confident that the suggested changes to the manuscript will respond to their concerns.

In particular, in the new version of the manuscript, we will discuss the importance of using a high resolution for detecting spatial gradients involving signs and the statistical significance of the projected changes, and will improve the comparison with the existing literature and the followed methodology. Moreover, we will justify and discuss the choice of the boundary conditions at the Gibraltar strait.

**Specific comments:**

**Biogeochemical spin-up:**

Some important information about the spin-up is missing for biogeochemistry. Please specify how you did the spinup and how long it runs before being stable. It is briefly mentioned in section 3.1 but this should be presented before. We thank the Reviewer for raising this point. We realized that this important information was missing in the manuscript. All the simulations discussed in the manuscript use as initial conditions the resulting final fields from a spin-up simulation. The latter starts on January, 1st 2005, following a spin-up of 100 years which reproduces repeatedly an average present condition corresponding to the 2005–2014 period looped for both physical forcing and river discharge. In the spin-up simulation, after the first 40-50 years the signal is almost stable in the first 1000 m and below 2000m while it takes a little longer to adjust between 1000-2000 m.

In order to better address the Reviewer's comment, we propose to include the following sentence:

"Finally, all the simulations discussed in the next sections, use as initial conditions the resulting final fields from a run that started in January, 1st 2005 following a spin-up of 100 years made with a loop over the 2005–2014 period for the physical forcing, the river nutrient discharge and atmospheric forcing (nutrient deposition and CO2 air value)."

**Boundary conditions:**

The boundary conditions are kept constant except for the DIC. Why all the variables of the carbonate cycle aren't treated the same way? Furthermore, did you use boundary conditions from a global RCP 8.5 for the MED RCP 8.5 simulation and a global RCP 4.5 for the MED RCP 4.5 simulation? Overall, this needs to be discussed as it could change the DIC / ALK ratio at Gibraltar Strait that might be important for pH variations in the MED (in conclusion). Please, specify how the DIC concentration evolves at Gibraltar Strait. A table representing nutrients and carbonate system variables for Gibraltar Strait and rivers will be appreciated.

We thank the Reviewer for raising this point that was not thoroughly discussed in the manuscript. Setting the biogeochemical BCs at the Gibraltar Strait is not straightforward because no-verifiable and anomalous simulated tendencies outside the Strait can result in spurious trends within the basin.

We considered the temporal evolution of the carbonate system and dissolved nutrients concentration in the Atlantic area outside the Gibraltar Strait during the 21st century in the CMCC-CESM global simulation (Vichi et al., 2011). However, after a preliminary analysis, the projected changes simulated at the Gibraltar Strait for nutrients have not been used as boundary conditions due to the anomalous nitrate/phosphate ratio observed. Because of that we prefer to keep the profile corresponding to the present conditions. Moreover, the analysis of the temporal evolution during the 21st century of DIC and total alkalinity in the Atlantic area outside the Gibraltar Strait, showed a continuous increase in the DIC concentration (more than 4% with respect to the present) and stable value of total alkalinity (the variation is around zero in the 21st century). We considered these signals reasonable and, because of that, we kept the CMCC-CESM simulated vertical profiles of DIC at the Gibraltar Strait while the total alkalinity was kept constant throughout the simulation. We acknowledge that we should have used two different BCs for the BGC scenarios. However, besides the lack of availability of a consistent RCP4.5 (w.r.t. the used RCP8.5) global scenario, we decided to test the impact of the different scenario atmospheric forcings (i.e., atmospheric pCO2 and physical dynamics) keeping the BC constant. Thus, our analysis revealed the impact of different atmospheric forcing but, possibly, could have underestimated a potential difference between the two scenarios. The manuscript will be modified taking into account the Reviewer's concern. In particular we propose to write in section 2.3:

"Boundary conditions are adopted to represent the external supply of biogeochemical tracers and properties from the Gibraltar Strait and the Mediterranean rivers into the basin. The exchanges of nutrients and other biogeochemical tracers and properties in the Gibraltar Strait are achieved by relaxing the 3D fields in the Atlantic zone (Fig. 1) to average vertical profiles which, for dissolved oxygen, phosphate, nitrate and silicate, refer to Salon et al. (2019), while total alkalinity is based on what was described in Cossarini et al. (2015). These profiles do not consider a seasonal cycle or a future temporal evolution, with DIC as the only exception, which is prescribed from a global ocean-climate simulation under RCP8.5 emission scenario performed within the framework of the CMIP5 project (Coupled Model Intercomparison Project Phase 5; Taylor et al., 2012) and based on the CMCC-CESM modeling system (CMCC-Coupled Earth System Model; Vichi et al., 2011). The reasons for these choices rely on: (i) anomalous values observed in N:P ratio under the RCP8.5 emission scenario, (ii) negligible variation, under emission scenario RCP8.5, of the total alkalinity along the 21st century, (iii) lack of a consistent RCP4.5 scenario, (iv) the possibility, using the same conditions at the Atlantic boundary, to test the impacts of the different atmospheric and ocean forcings."

Moreover, in the Discussion and Conclusions section we will acknowledge that:

"Moreover, the use of the same Atlantic boundary conditions for the two scenarios (section 2.3) could have led to an underestimation of a potential difference between the two scenarios in the areas most influenced by the boundary (e.g., Alboran and Southern Western Mediterranean)"

Finally, we acknowledge that the Gibraltar strait is an interesting area to be investigated. However, the paper provides evidence of basin wide signals without focusing on specific areas due to the paper length limit and some plots of evolution on some specific areas, including Alboran Sea, are already in Fig. S4-7. If not further required by the Reviewer, we would prefer not to include other figures and tables due to the already high amount of materials discussed in the manuscript and just to direct the potential Readers to the provided references.

**Periods selection PRESENT MID and FAR:**

A PRESENT period of 15 years was chosen and compared with MID and FAR periods of 20 years. How this could impact your results? Why don't you choose 20 years as well for the PRESENT period? During PRESENT period (2005-2020), the simulations are already in scenarios mode, as the RCP scenarios start in 2005. Therefore, the PRESENT period encompasses change linked to climate change which could lead to bias for evaluating the effect induced by climate change. Indeed, major differences between the 2 simulations considering the 2 RCP scenarios may appear during the PRESENT period, as observed for zonal stream function on your figure 4. Usually, the reference period is chosen among the hindcast part of the simulation to avoid those issues (Richon et al. 2019; Pages et al. 2020b, and others). Did you run a hindcast part before the scenarios? Please, at least, discuss your choice of the reference period and how it may affect your results.

We thank the Reviewer for highlighting this point, since we realize that also this information was missing in the text. We are aware that, in the CMIP5 simulations protocol, there is an historical simulation which covers the period 1950-2005 and then a scenario simulation which spans over the period 2006-2100. The choice to consider the period 2005-2020 as present climate for the validation relies on the idea to use more advanced datasets such as the CMEMS reanalysis (Teruzzi et al., 2019; Cossarini et al., 2021) and satellite Chl-*a* dataset (Colella et al., 2016) to evaluate the performance of the

modeling system. Being that both the datasets cover a period spanning from 1999 to 2020, in order to avoid the overlapping between historical and scenario we limited the evaluation to the period 2005-2020. We agree with the Reviewer that the choice of the period could introduce a potential error in our analysis due to the fact that we are already in "scenario mode". However, we tested the statistical significance of the differences between two scenarios in the period 2005-2020 for temperature, salinity and current speed fields and we found that the differences are not statistically significant over most of the basin (with only exception of small areas in the southern Ionian, Adriatic Sea and Levantine basin) and anyhow lower than the climate change signal. We do not expect that the choice of the period of different length as PRESENT and MID-FUTURE / FAR-FUTURE significantly impacts our findings. However, in order to address the Reviewer's comment, we propose to add the following sentence:

"The period 2005-2020 has been chosen as reference (also in the forthcoming validation) due to: (i) the availability, after 2000, of more advanced satellite and assimilated datasets to evaluate the biogeochemistry of the basin, (ii) to avoid the overlapping between historical and scenario part of the simulations (with the latter starting in 2005), (iii) the observed differences during this period between the two scenarios have been found not statistically significant over most of the basin (not shown)."

**Statistical significance:**

There is no indication in the text about the significance of the differences obtained for comparison between the MID and FAR periods. The word 'significant' is used but without statistics. In order to evaluate whether the numeric differences are substantial, it is necessary to calculate some parameters such as the t-student (See line 517 as an example) and to indicate if the variations are significant in the text and in the figures.

We thank the Reviewer for raising this point. We propose to redraw all the figures following their suggestion. In particular, we propose to assess the statistical significance of the differences in each point of the domain between PRESENT and MID-FUTURE and between PRESENT and FAR-FUTURE using a Mann-Whitney test with p

---

## Author Comment (AC3)

**Author responses to Reviewer#1 comments for the manuscript: "Acidification, deoxygenation, nutrient and biomasses decline in a warming Mediterranean Sea"**

**February, 18th 2022**

The authors projected the climate change-related impacts in the marine ecosystems of the Mediterranean Sea in the middle and at the end of the 21st century using an offline coupling model combining the physical model MFS16 and the transport-reaction model OGSTM-BFM, under emission scenarios RCP4.5 and RCP8.5, focusing on the middle and the end of 21st century. Projected changes are presented for temperature, salinity, dissolved nutrients and oxygen, net primary production, respiration, organic matter, plankton and bacterial biomass, particulate organic matter, and biogeochemical parameters (DIC, pH).

**The paper provides interesting projections in a changing Mediterranean Sea that is already under multiple pressures.**

We thank the Reviewer #1 for their positive feedback and for providing detailed comments and suggestions, which will be considered to improve the manuscript. Reviewer's comments are in bold, authors' responses are in normal font, italicized where they quote the proposed changes to the manuscript.

**Major comments:**

1) P3, L75: No, not "all" the modelling studies focused on high emissions scenarios. For example, there is Benedetti et al. (2018) who used A2, A1B and B1, and Goyet et al. (2016) who used B1 and A1F1.

- Benedetti, F., Guilhaumon, F., Adloff, F. and Ayata, S.-D. (2018), Investigating uncertainties in zooplankton composition shifts under climate change scenarios in the Mediterranean Sea. Ecography, 41: 345-360. https://doi.org/10.1111/ecog.02434
- Goyet, C., Hassoun, A., Gemayel, E., Touratier, F., Abboud-Abi Saab, M. and Guglielmi, V., 2016. Thermodynamic forecasts of the mediterranean sea acidification. Mediterranean Marine Science, 17(2), pp.508-518. Thermodynamic Forecasts of the Mediterranean Sea Acidification | GOYET | Mediterranean Marine Science (ekt.gr).

Goyet et al. (2016) is the only modelling study that it is projecting carbonate system parameters in the Mediterranean Sea so far, and the one used in MedECC (2020; cited by the authors to tackle OA projections in the Mediterranean). Yet, it is not mentioned at all in this work. Please check this study out and try to compare your results with theirs.

We thank the Reviewer for pointing out this error in the sentence. We propose to include and discuss the suggested references in the manuscript. In the introduction we will state:

"An assessment of the effects of climate change on the biogeochemistry and marine ecosystem dynamics of the Mediterranean Sea has been considered in a certain number of studies based on different emission scenarios".

"Benedetti et al. (2018), using environmental niche models and considering six physical simulations based on different emission scenarios (A2, A2-F, A2-RF, A2-ARF, A1B-ARF, B1-ARF; Adloff et al., 2015), projected, in response to climate change, a loss of copepods diversity throughout most of the surface of the Mediterranean Sea."

Goyet et al. (2016) will be discussed in the Discussions and Conclusions section where it will be stated that:

"The overall accumulation of  $CO_2$  in the basin resulted in an acidification of the Mediterranean water with a decrease in pH of approximately 0.23 units, which is slightly lower than the 0.3 projected on a global scale (Kwiatkowski et al., 2020) and lower than the value provided in Goyet et al. (2016), who projected, using thermodynamic equations of the  $CO_2$ /carbonate system chemical equilibrium in seawater, a variation of 0.45 pH units in the basin under the worst SRES case scenario (0.25 pH units in the most optimistic SRES scenario). However, this last estimate probably tends to overestimate the future acidification of the basin, as it does not consider the decrease in the exchanges and the penetration of  $CO_2$  across the ocean-atmosphere interface due to the warming of the water column (MedECC, 2020)."

2) P3, L78-79: The authors mentioned that Moullec et al. (2019), under RCP8.5 emission scenario, found an increase in both phytoplankton biomass and net primary production by the end of the 21st century. However, this pattern is not homogenous in the Mediterranean since Moullec et al. (2019) have also highlighted a difference between the Eastern and Western basins with an increase in the first and a decrease in the second. Please edit accordingly.

Agreed. The paragraph will be reformulated as follows:

"On the other hand, Moullec et al. (2019), under RCP8.5 emission scenario, found an increase/decrease in both phytoplankton biomass and net primary production by the end of the 21st century in the Eastern/Western Mediterranean Sea."

P4, L111-113: In addition to the BOUM mesoscale experiments working on relating eddies with biogeochemical changes (BG - Influence of anticyclonic eddies on the Biogeochemistry from the Oligotrophic to the Ultraoligotrophic Mediterranean (BOUM cruise) (copernicus.org)), there are actually many modelling studies, for example:

- Ramirez-Romero E, Jordà G, Amores A, Kay S, Segura-Noguera M, Macias DM, Maynou F, Sabatés A and Catalán IA (2020) Assessment of the Skill of Coupled Physical–Biogeochemical Models in the NW Mediterranean. Front. Mar. Sci. 7:497. doi: 10.3389/fmars.2020.00497
- Guyennon, A., Baklouti, M., Diaz, F., Palmieri, J., Beuvier, J., Lebaupin-Brossier, C., Arsouze, T., Béranger, K., Dutay, J.-C., and Moutin, T.: New insights into the organic carbon export in the Mediterranean Sea from 3-D modeling, Biogeosciences, 12, 7025–7046, https://doi.org/10.5194/bg-12-7025-2015, 2015.

• Herrmann, M., Somot, S., Sevault, F., Estournel, C., and Déqué, M. (2008), Modeling the deep convection in the northwestern Mediterranean Sea using an eddy-permitting and an eddy-resolving model: Case study of winter 1986–1987, J. Geophys. Res., 113, C04011, doi:10.1029/2006JC003991.

**Therefore, I would suggest to re-write this paragraph.**

We agree with the Reviewer that, before our work, there have already been other observational/modeling efforts to resolve the eddies dynamics in the Mediterranean Sea and its impacts on the biogeochemistry. However, our work has two specific differences with respect to the available literature: i) the works listed by the Reviewer, although being eddy-resolving, focus either on limited areas of the Mediterranean Sea or on a single variable/physical process; (ii) they are hindcasts and do not provide projections for the biogeochemical variables under different emission scenarios. In order to address the Reviewer's comment, we propose to reformulate the paragraph as follows:

"These considerations emphasize the importance of providing eddy-resolving future projections of the Mediterranean Sea biogeochemistry that further extends the analysis of the climate change-related impacts in the marine ecosystems of the basin under different emission scenarios. In fact, although in the recent period observational and modeling studies have been carried out to further highlight the importance of the mesoscale dynamics in the physical and biogeochemical state of specific areas of the Mediterranean Sea (e.g. Hermann et al., 2008; Moutin and Prieur, 2012; Guyennon et al., 2015; Ramirez-Romero et al., 2020), long-term eddy-resolving biogeochemical projections under different emission scenarios, to the best of the authors' knowledge, have not been analyzed so far in the region. Such projections might be used in future studies specifically focused on the analysis of climate change impact on specific organisms, habitats and/or local areas."

P8, L263-266: To characterize the spatial distribution and the variability of anomalies, the authors considered their horizontally averages in each sub-basin in the Western Mediterranean (WMED=(ALB+SWM+NWM+TYR)/4) and in only two sub-basins of the Eastern Mediterranean (EMED=(ION+LEV)/2). Why did you exclude the Adriatic and the Aegean Sub-basins here?

We thank the Reviewer for pointing this issue, because we realized that this explanation was missing in the manuscript. Here we followed the approach already adopted in the other works (for example Lazzari et al., 2012; 2016; Di Biagio et al., 2019; Reale et al., 2020 a,b) where the characteristics of both Adriatic and Aegean Sea (for example the paramount importance of riverine inputs or Dardanelles straits in the biogeochemical dynamics of both basins) are considered such peculiar to make both basins separate with respect to the Eastern Mediterranean. For this reason, they are not included in our averages.

In order to address the Reviewer's comment, we will modify the sentence as follows:

"Horizontally spatial averages are computed considering the sub-basins defined in Fig. 1, the whole Mediterranean basin and two macro-areas: Western Mediterranean (WMED which includes ALB, SWM, NWM, TYR) and the Eastern Mediterranean (EMED which includes ION and LEV). The Adriatic and Ionian Sea are not usually considered part of the Eastern Mediterranean due to the importance of local forcing, such as riverine load, in shaping the variability of the biogeochemical dynamics in two sub-basins. Because of that, following the approach already adopted in previous works (Lazzari et al., 2012; 2016; Di Biagio et al., 2019; Reale et al., 2020 a,b) they are not considered in the spatial averages related to WMED and EMED."

P9-10, L306-315 & Fig. 2: The authors mentioned that mean simulated values in the first 0-200 m are quite realistic in all the variables, and that biases started to show at 600 m depth. However, the vertical profiles show such discrepancies between CTRL average profiles and observational data (EMODnet) even in shallower depths, i.e. less than 50 m for phosphate in the WMed., surface waters for nitrate in the WMed., greater than 200 m for oxygen, and so much general biases in pH. Could you please elaborate more on this?

We thank the Reviewer for raising this point and pointing out this lack in the manuscript. We agree with the Reviewer that the model clearly shows some underestimation/overestimation in some of the simulated biogeochemical variables that have not been thoroughly discussed. On the other hand, our validation shows that the main biogeochemical characteristics of the basin such as presence and features of the Deep Chlorophyll Maximum (DCM), nutricline deepening between Western and Eastern basin, low nutrient concentration at the surface, vertical profiles of DIC, spatial distribution of total alkalinity and pH are well simulated. Thus, without presuming to oversell our results, we believe that our modeling tool is fairly good to be used for scenario simulations, also considering the levels of validation of other scenario simulations published in literature (see for example Richon et al., 2019 and Solidoro et al., 2022).

We propose to further elaborate the paragraph by listing the major biases in variables and layers. More specifically we will rewrite the paragraph as follows:

"Figure 2 also shows the average vertical profiles, computed for the entire, Western and Eastern Mediterranean basin, of Chl-a (c), PO4 (d), NO3 (e), dissolved oxygen (f), DIC (g), pH (h) and total alkalinity (i) in the CTRL compared with the recent CMEMS reanalysis (only for Chl-a and pH, Teruzzi et al., 2021) and EMODnet datasets (European Marine Observation and Data Network; Buga et al., 2018). The model captures the DCM location, the west-east trophic gradient in the basin, the nutricline depth deepening between Western and Eastern basin and the low nutrient surface concentrations. Mean simulated values in the first 0-200 m are fairly realistic for almost all the variables, with correlation coefficients between observations and model larger than 0.93. At the same time, between 100 and 300 m the CTRL overestimates (underestimates) the PO4 concentration (pH) of about 50% (1%), and below 200 m it overestimates (underestimates) the dissolved oxygen (NO3) of about 15% (20%).

In general, these biases in the initial conditions come from the spin-up simulation, that allows the largest part of the model drift to be removed. Biases are still present in both the CTRL and scenario simulations while the eventually still-present drifts in the CTRL are by far lower than the climate signal.

To summarize, although the model shows some deficiencies in simulating the vertical distribution of some biogeochemical variables, the main biogeochemical features of the basin are very well simulated and thus, MFS16-OGSTM-BFM can be used to investigate the evolution of the Mediterranean biogeochemistry under different emission scenarios."

**P11, L326: Could you explain in the text the depth classification adopted in this study: 0-100 m and 200-600 m?**

Agreed. The sentence will be modified as follows:

"Mean temperature and salinity evolution between 0-100 m and 200-600 m in the 2005-2099 period under the RCP4.5 and RCP8.5 scenarios in the whole Mediterranean Sea and in the Western and Eastern basins, are shown in Fig. 3. As for the biogeochemical variables, these depths have been chosen as they represent the location of MAW and LIW, respectively".

**P12, L342-358: Is it possible to check if those differences are significant or not?**

Following also the Reviewer#2's suggestion we propose to extensively redraw the figures including also an assessment of the statistical significance of the observed difference using the Mann-Whitney test with p

---

## Author Response (AR1)

**Author responses to Editor comments for the manuscript: "Acidification, deoxygenation, nutrient and biomasses decline in a warming Mediterranean Sea" April, 2022**

We thank the Editor for their positive feedback and for providing detailed comments and suggestions, that helped us to improve the manuscript. Editor's comments are in bold, authors' responses are in normal font, italicized where they quote the changes in the manuscript.

I hope this comment does not come too late. I refer to figure 18 panels e-l. It is inappropriate to express changes in pH in percent. It should be percent of the hydrogen ion concentration. Or, perhaps better, show the changes in pH units. See: Fassbender A. J., Orr J. C. & Dickson A. G., 2021. Technical note: interpreting pH changes. Biogeosciences 18:1407-1415.

We thank the Editor for pointing out the error in Figure 18. It has been redrawn following the Editor's suggestion. Here we show the "new" Figure 18 with its relative caption:

---

## Referee Report (RR1)

**Comments on the revised manuscript "Acidification, deoxygenation, nutrient and biomasses decline in a warming Mediterranean Sea"**

I would like to thank the authors for the quality of their reviewed manuscript. The revised version of the manuscript is now much clearer, all my questions have been addressed, the modifications have been made to the manuscript. The simulation protocol has been re-written and other technical information (spin-up, boundary conditions…) have been added. The figures and their captions also significantly improved. The discussions have been significantly changed and now propose a more rigorous comparison with the existing literature and emphasize more the importance of using a high resolution. Finally, it also focuses more on the differences between the RCP4.5 and RCP8.5 scenarios.

I believe that this paper is now ready to be published and that this work is a significant step forward toward a better understanding of the effects of climate change on the Mediterranean Sea. I list below a few minor changes that should be made before publication.

**Minor remarks :**

Line 83: "CO2" abbreviation hasn't been defined before

Line 214: You introduce ALK and you don't use it on line 215 or line 227, be careful to be consistent with the abbreviations.

Line 341: "the resolution to 1/24 degree"; Change "degree" for the symbol to be consistent with the lines above.

Line 346: "chlorophyll-a (Chl-a)"; You could introduce this before and remove the chlorophyll-a, it would be more consistent. See line 943 as well.

*Bibliography:* Some references end with a dot "xxx 2019." and others end with the date without a dot, see for example line 1122, 1141.

*Figures :*

The figures need to be carefully checked. I will list here figure by figure what needs to be corrected. A general remark is that the font size between all the figures needs to be coherent.

Fig.2: h) the unit is false for pH

    g&i ) space between mu and mol

Fig.15: space between mu and mol for DIC unit

Fig.16: c & d) you use d-1 here, but usually you use year-1 so use day-1

Fig.17: a) end bracket is missing for concentration

Fig.S4: Yaxis title "MLD year [m]" why  year ?

Fig S5: you use Mmol but if I am not mistaken in the manuscript you use mmol please correct this (as well in the other figures)

Sp1: I like this table that is a great improvement, the caption however could be clearer.

The values are averaged spatially and between 0-100 m and 200-600 m?

Adding the std could be nice.

---

## Author Response (AR2)

**Author responses to Editor comments for the manuscript:**

**"Acidification, deoxygenation, nutrient and biomasses decline in a warming Mediterranean Sea"**

**May, 2022**

We thank Dr. Abed El Rahman Hassoun and the Anonymous Reviewer#2 for their positive feedback and for providing detailed comments and suggestions that helped us to further improve the manuscript. Reviewer's comments are in bold, authors' responses are in normal font, italicized where they quote the changes in the manuscript.

**Reply to Reviewer#1 (Dr. Abed El Rahman Hassoun)**

**1-For the new table of projections, it would be very helpful to also add the projected anomalies.** We thank the Reviewer for his comment in the previous revision related to the need for a table in the manuscript. However, the anomalies are already discussed in the manuscript and we think that adding these anomalies to the table SP1 where the absolute values are already shown would make this information just redundant. In any case, in order to handle the request of the other Reviewer the table was further extended by reporting the temporal standard deviation values associated with each variable.

**2-For the comment of "P8, L263-266", although it is adopted in other published studies, I do not agree to remove the Adriatic and Aegean Sub-basins, since they both, with the remaining sub-basins, give to the Eastern Mediterranean basin its peculiarity. Also, the main water masses are circulating in all Eastern Med. Sub-basins and it is not logical to separate them as if they are not connected. For example, high alkalinity waters coming from the Black Sea, through the Aegean, towards the Levantine Sub-basin, are the reason why there is high AT/buffering capacity in the latter sub-basin compared to other Mediterranean areas, etc. BUT, I will not request to change this in the current manuscript, just wanted to reflect my point of view.**

We thank the Reviewer for this comment. We think that including/excluding Adriatic and Aegean Seas into/from the Eastern Mediterranean Sea should be decided/justified on the basis of the scope of the analysis. In our specific case, we decided to exclude them for sake of consistency and for allowing a quick comparison of the results of our numerical simulations with the previous works discussing the biogeochemical dynamical of the Mediterranean Sea (Lazzari et al., 2012; 2016; Di

Biagio et al., 2019; Reale et al., 2020 a,b; Cossarini et al, 2021). In any case, the peculiarity of the Adriatic and Aegean Sea is shown in the maps and tables of the manuscript.

**3- For the comment of "P11, L326", thanks for adding a reference at the end of the new sentence.**

On our side, we thank the Reviewer for this suggestion.

**4- In lines 332-334: please add 'spatio-temporal' instead of 'spatial-temporal'.**
Done

**5- For references in lines 917-920, the reference "Hassoun et al., 2019" is not correct. I think you mean here "Hassoun et al., 2015". Hassoun, A.E.R., Gemayel, E., Krasakopoulou, E., Goyet, C., Abboud-Abi Saab, M., Guglielmi, V., Touratier, F. and Falco, C., 2015. Acidification of the Mediterranean Sea from anthropogenic carbon penetration. Deep Sea Research Part I: Oceanographic Research Papers, 102, pp.1-15.**

Yes, we were mistaken. The citation has been changed accordingly.

**Reply to Reviewer#2**

**Minor remarks:**
**Line 83: "CO2" abbreviation hasn't been defined before**
Done
**Line 214: You introduce ALK and you don't use it on line 215 or line 227, be careful to be consistent with the abbreviations.**
Agreed. The text has been modified accordingly
**Line 341: "the resolution to 1/24 degree"; Change "degree" for the symbol to be consistent with the lines above.**
Agreed. The text has been modified accordingly
**Line 346: "chlorophyll-a (Chl-a)"; You could introduce this before and remove the chlorophyll-a, it would be more consistent. See line 943 as well.**
We thank the Reviewer for the suggestion. The text has been modified accordingly

**Bibliography: Some references end with a dot "xxx 2019." and others end with the date without a dot, see for example line 1122, 1141.**

We thank the Reviewer for spotting this inconsistency in the references. The text has been modified accordingly adding a dot at the end of each reference.

**Figures:**

**The figures need to be carefully checked. I will list here figure by figure what needs to be corrected. A general remark is that the font size between all the figures needs to be coherent.**

**Fig.2: h) the unit is false for pH**

**g&i ) space between mu and mol**

**Fig.15: space between mu and mol for DIC unit**

**Fig.16: c & d) you use d-1 here, but usually you use year-1 so use day-1**

**Fig.17: a) end bracket is missing for concentration**

We thank the Reviewer for spotting the errors in the figures that have been corrected accordingly

**Fig.S4: Yaxis title "MLD year [m]" why year?**

Unfortunately, there was a typo in the uploaded figure, which has been modified by removing the word "year" in the Y-axis title.

**Fig S5: you use Mmol but if I am not mistaken in the manuscript you use mmol please correct this (as well in the other figures)**

The values reported in Fig.S5-S6-S7 represents the mass balance of phosphate and nitrate through the selected straits and it is measured in Megamoles year$^{-1}$(Mmol year$^{-1}$). In the manuscript we reported the absolute concentration of nitrate and phosphate in millimole/m$^{-3}$ (mmol/m$^{-3}$).

**Sp1: I like this table that is a great improvement, the caption however could be clearer. The values are averaged spatially and between 0-100 m and 200-600 m? Adding the std could be nice.**

Agreed. Moreover, we thank the Reviewer for suggesting the addition of the standard deviation to the table. The table has been modified including the temporal standard deviation of the "unbiased scenario" values. Both table and caption have been modified as follows:

|  |  | RCP4.5 | | | RCP8.5 | | |
|---|---|---|---|---|---|---|---|
|  |  | PRESENT | MID-FUTURE | FAR-FUTURE | PRESENT | MID-FUTURE | FAR-FUTURE |
| **Seawater Temperature (°C)** | | | | | | | |
| WMED | 0-100 | 16.3±0.3 | **16.8±0.3** | **17.5±0.2** | 16.4±0.3 | **17.2±0.4** | **19.0±0.3** |
| | 200-600 | 13.9±0.1 | **14.9±0.1** | **15.6±0.1** | 14.0±0.1 | **15.2±0.1** | **16.6±0.2** |
| EMED | 0-100 | 18.2±0.2 | **18.9±0.3** | **19.8±0.2** | 18.4± 0.2 | **19.5±0.5** | **21.7±0.4** |
| | 200-600 | 14.5±0.1 | **15.0±0.1** | **15.8±0.0** | 14.6±0.1 | **15.3±0.1** | **16.8±0.2** |
| **Seawater Salinity (-)** | | | | | | | |
| WMED | 0-100 | 37.4±0.1 | **36.9±0.1** | **37.0±0.1** | 37.4±0.1 | **36.9±0.1** | **37.0±0.1** |
| | 200-600 | 38.6±0.0 | **38.8±0.0** | **38.7±0.0** | 38.6±0.0 | 38.9±0.0 | **39.0±0.0** |
| EMED | 0-100 | 38.6±0.1 | **38.3±0.1** | **38.5±0.1** | 38.6±0.1 | **38.4±0.1** | **38.8±0.1** |
| | 200-600 | 38.9±0.0 | 38.9±0.0 | **38.9±0.0** | 38.9±0.0 | **39.0±0.0** | **39.1±0.1** |
| **PO$_4$ (mmol m$^{-3}$)** | | | | | | | |
| WMED | 0-100 | 0.14±0.01 | **0.13±0.00** | 0.14±0.01 | 0.14±0.00 | **0.13±0.00** | **0.13±0.00** |
| | 200-600 | 0.29±0.00 | **0.30±0.01** | **0.28±0.00** | 0.30±0.00 | **0.29±0.01** | **0.29±0.00** |
| EMED | 0-100 | 0.03±0.00 | **0.03±0.00** | **0.03±0.00** | 0.03±0.00 | **0.03±0.00** | **0.02±0.00** |
| | 200-600 | 0.18±0.00 | **0.18±0.00** | **0.17±0.00** | 0.17±0.00 | 0.17±0.00 | 0.17±0.00 |
| **NO$_3$ (mmol m$^{-3}$)** | | | | | | | |
| WMED | 0-100 | 1.0±0.1 | 1.0±0.1 | 1.0±0.1 | 1.0±0.1 | **0.9±0.0** | **0.9±0.0** |
| | 200-600 | 4.6±0.0 | **4.7±0.1** | **4.5±0.0** | 4.7±0.0 | **4.7±0.1** | **4.6±0.0** |
| EMED | 0-100 | 0.2±0.0 | **0.1±0.0** | **0.1±0.0** | 0.2±0.0 | **0.1±0.0** | **0.1±0.0** |
| | 200-600 | 3.1±0.0 | **3.0±0.0** | **3.0±0.0** | 3.0±0.0 | 3.0±0.0 | 3.0±0.0 |
| **Dissolved Oxygen (mmol m$^{-3}$)** | | | | | | | |
| WMED | 0-100 | 237±1 | **233±1** | **231±1** | 235±1 | **230±2** | **223±1** |
| | 200-600 | 213±1 | **207±1** | **204±1** | 211±0 | **205±0** | **196±2** |
| EMED | 0-100 | 234±1 | **232±1** | **228±1** | 233±1 | **229±2** | **219±1** |
| | 200-600 | 219±1 | **217±0** | **213±0** | 220±1 | **217±1** | **208±2** |

| Phytoplankton biomass (mg m$^{-3}$) | | | | | | | |
|---|---|---|---|---|---|---|---|
| WMED | 0-100 | 12.9±0.5 | **11.8±0.6** | **11.9±0.6** | 12.7±0.6 | **11.5±0.6** | **10.4±0.5** |
| EMED | 0-100 | 7.5±0.5 | **6.3±0.5** | **6.2±0.4** | 7.0±0.4 | **5.7±0.5** | **4.8±0.3** |
| **Zooplankton biomass (mg m$^{-3}$)** | | | | | | | |
| WMED | 0-100 | 14.5±0.3 | **14.0±0.4** | **14.2±0.4** | 14.6±0.4 | **14.0±0.4** | **13.7±0.5** |
| EMED | 0-100 | 11.7±0.3 | **10.9±0.3** | **10.9±0.3** | 11.4±0.2 | **10.5±0.4** | **9.8±0.4** |
| **Integrated net primary production (gC m$^{-2}$ year$^{-1}$)** | | | | | | | |
| WMED | 0-200 | 135±6 | 134±3 | **144±4** | 137±5 | **139±3** | **156±8** |
| EMED | 0-200 | 140±5 | **136±3** | **149±4** | 137±4 | **139±3** | **162±6** |
| **Dissolved Inorganic carbon (µmol kg$^{-1}$)** | | | | | | | |
| WMED | 0-100 | 2276±7 | **2297±4** | **2322±4** | 2270±2 | **2301±8** | **2375±12** |
| | 200-600 | 2373±0 | **2404±7** | **2447±4** | 2375±1 | **2404±11** | **2495±14** |
| EMED | 0-100 | 2325±4 | **2363±12** | **2400±4** | 2318±3 | **2372±14** | **2484±15** |
| | 200-600 | 2382±2 | **2410±6** | **2452±3** | 2379±2 | **2412±10** | **2505±15** |
| **pH (-)** | | | | | | | |
| WMED | 0-100 | 8.07±0.00 | **8.03±0.01** | **8.00±0.00** | 8.07±0.01 | **8.01±0.02** | **7.88±0.02** |
| | 200-600 | 8.08±0.00 | **8.05±0.00** | **8.00±0.00** | 8.09±0.00 | **8.04±0.01** | **7.92±0.02** |
| EMED | 0-100 | 8.09±0.00 | **8.03±0.01** | **7.99±0.00** | 8.09±0.00 | **8.00±0.02** | **7.84±0.02** |
| | 200-600 | 8.1±0.00 | **8.06±0.00** | **8.02±0.00** | 8.10±0.01 | **8.05±0.01** | **7.92±0.02** |

*"Table SP1 "Unbiased scenario" values for seawater temperature, salinity, dissolved phosphate, nitrate and oxygen concentrations, Phytoplankton and Zooplankton biomass at surface, vertically integrated net primary production, Dissolved Inorganic Carbon and pH in the PRESENT (2005-2020), MID-FUTURE (2040-2059) and FAR-FUTURE (2080-2099) time windows. Averages and temporal standard deviations are computed considering the timeseries of annual means for the Western (WMED) and Eastern (EMED) Mediterranean Sea for the layers 0-100 m and 200-600 m. Bold format indicates significant differences of the future averages from the values of the PRESENT period according to a Mann-Whitney test with p<0.05."*